# The p97–Ataxin 3 complex regulates homeostasis of the DNA damage response E3 ubiquitin ligase RNF8

Abhay Narayan Singh[1,§], Judith Oehler[1,§,†], Ignacio Torrecilla[1], Susan Kilgas[1], Shudong Li[1], Bruno Vaz[1], Claire Guérillon[2], John Fielden[1], Esperanza Hernandez-Carralero[3,4], Elisa Cabrera[3,4], Iain DC Tullis[1], Mayura Meerang[5,‡], Paul R Barber[1], Raimundo Freire[3,4,6], Jason Parsons[7], Borivoj Vojnovic[1], Anne E Kiltie[1] [iD], Niels Mailand[2] [iD] & Kristijan Ramadan[1,*] [iD]

## Abstract

The E3 ubiquitin ligase RNF8 (RING finger protein 8) is a pivotal enzyme for DNA repair. However, RNF8 hyper-accumulation is tumour-promoting and positively correlates with genome instability, cancer cell invasion, metastasis and poor patient prognosis. Very little is known about the mechanisms regulating RNF8 homeostasis to preserve genome stability. Here, we identify the cellular machinery, composed of the p97/VCP ubiquitin-dependent unfoldase/segregase and the Ataxin 3 (ATX3) deubiquitinase, which together form a physical and functional complex with RNF8 to regulate its proteasome-dependent homeostasis under physiological conditions. Under genotoxic stress, when RNF8 is rapidly recruited to sites of DNA lesions, the p97–ATX3 machinery stimulates the extraction of RNF8 from chromatin to balance DNA repair pathway choice and promote cell survival after ionising radiation (IR). Inactivation of the p97–ATX3 complex affects the non-homologous end joining DNA repair pathway and hypersensitises human cancer cells to IR. We propose that the p97–ATX3 complex is the essential machinery for regulation of RNF8 homeostasis under both physiological and genotoxic conditions and that targeting ATX3 may be a promising strategy to radio-sensitise BRCA-deficient cancers.

**Keywords** Ataxin 3; DNA double-strand break repair; E3 ubiquitin ligase RNF8; genome stability; p97/VCP ATPase
**Subject Categories** DNA Replication & Repair; Post-translational Modifications & Proteolysis
**The EMBO Journal (2019) 38: e102361**

## Introduction

DNA double-strand breaks (DSBs) are the most deleterious DNA lesions and, if not accurately repaired, can lead to chromosomal aberrations, immunodeficiency, neurodegeneration, cancer, accelerated ageing and cell death (Friedberg *et al*, 2006; Jackson & Bartek, 2009; Panier & Durocher, 2013; Jeggo *et al*, 2016).

Mammalian cells execute two main DSB repair pathways: non-homologous end joining (NHEJ) and homologous recombination (HR), promoted by 53BP1 and BRCA1/BRCA2/Rad51, respectively (Chapman *et al*, 2012; Aparicio *et al*, 2014; Hustedt & Durocher, 2016). The coordination of DSB repair is under the strict control of various post-translational modifications (PTMs). These include ubiquitination generated by two E3 ubiquitin ligases RNF8 and RNF168 which has emerged as an essential PTM for DSB repair pathway choice and cell survival (Kolas *et al*, 2007; Mailand *et al*, 2007; Doil *et al*, 2009; Pinato *et al*, 2009; Stewart *et al*, 2009; Gatti *et al*, 2015).

RNF8 is the first E3 ubiquitin ligase recruited to sites of DSBs, where it ubiquitinates histone H1 with K63-ubiquitin-linked chains (K63-Ub) in cooperation with the E2 ubiquitin conjugating enzyme Ubc13 (Thorslund *et al*, 2015). RNF8-dependent H1/K63-Ub is essential to recruit the second E3 ubiquitin ligase RNF168. RNF168 further amplifies and spreads the ubiquitin signal onto the surrounding histones, H2A-type histones, that serve as docking sites for 53BP1 recruitment to chromatin to prevent excessive 5′-DNA end resection (the initial step of the HR pathway) and to promote NHEJ-mediated repair (Mattiroli *et al*, 2012; Chang *et al*, 2017; Dev *et al*, 2018; Noordermeer *et al*, 2018).

BRCA1 and Rad51, the essential enzymes for HR, are also recruited to sites of DSBs in a ubiquitin-dependent manner; however, their recruitment is strictly dependent on RNF8 and not RNF168 (Kim *et al*, 2007; Sobhian *et al*, 2007; Wang & Elledge, 2007; Sy *et al*, 2011;

1  Department of Oncology, Cancer Research UK/Medical Research Council Oxford Institute for Radiation Oncology, University of Oxford, Oxford, UK
2  Novo Nordisk Foundation Center for Protein Research, University of Copenhagen, Copenhagen, Denmark
3  Unidad de Investigación, Hospital Universitario de Canarias, La Laguna, Spain
4  Instituto de Tecnologías Biomédicas, Universidad de La Laguna, La Laguna, Spain
5  Institute of Pharmacology and Toxicology-Vetsuisse Faculty, University of Zurich, Zurich, Switzerland
6  Universidad Fernando Pessoa Canarias, Santa Maria de Guia, Spain
7  Department of Molecular and Clinical Cancer Medicine, Cancer Research Centre, University of Liverpool, Liverpool, UK
   *Corresponding author. Tel: +44 01865 617 349; E-mail: kristijan.ramadan@oncology.ox.ac.uk
   §These authors contributed equally to this work
   †Present address: Department of Biochemistry, University of Oxford, Oxford, UK
   ‡Present address: Department of Thoracic Surgery, University Hospital Zurich, Zurich, Switzerland

Munoz *et al*, 2012; Nakada *et al*, 2012; Watanabe *et al*, 2013; Zong *et al*, 2019), supporting the notion that HR is not absolutely dependent on the RNF168-mediated ubiquitin signalling cascade.

Separation of function between these two main DSB repair E3 ubiquitin ligases—RNF8 and RNF168—was further observed in RIDDLE syndrome patient cells bearing RNF168 mutations. These cells are radiosensitive due to their inability to recruit 53BP1 (NHEJ pathway) to sites of DSBs, while maintaining recruitment of BRCA1 and Rad51 (HR pathway) to the break sites (Stewart *et al*, 2007). This was also confirmed in the *Rnf168*-knockout mouse cells (Zong *et al*, 2019). Finally, structural and cell biological data further demonstrate an independent and direct role of RNF8 and its K63-Ub in BRCA1 recruitment to sites of DNA lesions (Hodge *et al*, 2016).

Altogether this suggests that the RNF168/Ubiquitin/53BP1 cascade is essential to mediate NHEJ and is functionally distinguished from the RNF8/K63-Ub/BRCA1/Rad51-governed HR repair pathway. Thus, to have balanced DSB repair, the levels of RNF8 and RNF168 have to be tightly regulated.

Unlike RNF168, whose protein expression is tightly regulated by two E3 ubiquitin ligases and a deubiquitinating enzyme (DUB), namely UBR5, TRIP12 and USP7, respectively (Gudjonsson *et al*, 2012; Zhu *et al*, 2015), how the homeostasis of RNF8 is regulated is not known. This is a key issue, not only for accurate DNA repair after genotoxic stress but also for genome stability under physiological conditions, since hyper-accumulation of RNF8 is directly linked to tumorigenesis and poor prognosis in breast cancer patients (Kuang *et al*, 2016; Lee *et al*, 2016a,b). Here, we present data demonstrating that RNF8 homeostasis is regulated by the ATPase p97 (p97)—also known as VCP unfoldase in humans—and its associated DUB Ataxin 3 (ATX3).

The AAA+ ATPase p97 is a homo-hexameric barrel-like protein, which forms multiple complexes and sub-complexes with its associated cofactors that confer p97 specificity to various substrates and cellular pathways (Meyer & Weihl, 2014; Buchberger *et al*, 2015). These p97 complexes bind (via cofactors) and process (via p97 ATPase activity) ubiquitinated substrates by either unfolding or segregating them from different cellular locations including chromatin (Vaz *et al*, 2013; Dantuma *et al*, 2014). In turn, ubiquitinated substrates remodelled by the p97 system are either presented to the proteasome for their degradation or recycled by DUBs (Meyer *et al*, 2012). Thus, the p97 system plays a central role in maintaining protein homeostasis.

Here, we report that p97 and ATX3 (Rao *et al*, 2017) form a constitutive physical complex with RNF8. The p97–ATX3 complex safeguards the soluble pool of RNF8 under physiological conditions. However, under genotoxic conditions, when RNF8 is rapidly recruited to sites of DNA lesions and orchestrates the DNA damage response, the p97–ATX3 complex facilitates RNF8 chromatin extraction to balance DSB repair pathway choice and improve cell survival to ionising radiation (IR). Our data reveal the p97–ATX3 complex as a crucial mediator of RNF8 regulation and identify a fundamental role of the p97–ATX3–RNF8 axis in promoting genome stability and resistance to IR.

## Results

### RNF8 is a ubiquitinated substrate of p97

To test the hypothesis that the p97 system regulates RNF8 homeostasis, we monitored the total pool of endogenous RNF8 in HeLa and HEK293 cells when either p97 was siRNA-depleted or its ATPase activity was blocked. The ATPase activity of p97 was blocked by either acute chemical inhibition (CB5083) or doxycycline (DOX)-inducible expression of the p97E578Q (p97EQ; dominant negative) variant. Inactivation of p97 by either approach led to increased total levels of endogenous RNF8 under physiological and IR-treated conditions (Fig 1A–F). Both chemically inhibited p97 and the p97EQ variant bind, but are unable to process, the substrates due to inactivation of p97 ATPase activity (Ye *et al*, 2003; Meerang *et al*, 2011; Fig 1G). Similarly, cycloheximide (CHX) chase experiments in p97-inactivated HEK293 cells, either by siRNA-mediated p97 depletion or by mild expression of p97EQ variant, demonstrated a marked delay in endogenous RNF8 degradation kinetics when compared to control cells (Figs 1H and I, and EV1A–C), suggesting that the rate of RNF8 degradation and its homeostasis are under the control of the p97 system and this effect is not cell type-specific.

As p97 unfolds and extracts predominantly ubiquitinated proteins from different cellular locations, we next asked if RNF8 is modified by ubiquitination. HEK293 cells co-transfected with Flag-RNF8 and HA-Ubiquitin were mock-treated or exposed to IR. The cells were biochemically fractionated into total soluble (cytosol and nucleosol) and chromatin pools, and Flag-RNF8 was subsequently immunopurified under denaturing conditions from both fractions to analyse its ubiquitination status (Fig EV1D). Ubiquitination of RNF8 was analysed using antibodies recognising either total ubiquitination (i.e. HA antibody) or specific ubiquitin linkages with a similar affinity (K48-Ub and K63-Ub; Fig EV1E). RNF8 was strongly ubiquitinated under both physiological and genotoxic conditions, and the K48-Ub appeared to be the main ubiquitin signal on both soluble and chromatin-bound RNF8 (Fig EV1D). Furthermore, RNF8 was hyper-ubiquitinated with K48-Ub in p97-depleted cells when compared to control cells (Fig 1J), further supporting a role for the p97 system in processing ubiquitinated RNF8.

To directly prove that RNF8 is a substrate of p97, we performed co-immunoprecipitation experiments and analysed the physical association between p97 and RNF8 *in vivo*. Flag-RNF8 and Strep-tagged p97 wild type (WT) or p97EQ were mildly co-expressed in HEK293 cells. The cells were then mock-treated or exposed to IR, and Strep-tagged p97 complexes were isolated from total cell extract over streptavidin beads under native and physiological salt (150 mM NaCl) conditions (Fig 1K and L). The p97-WT isolate precipitated equal amounts of RNF8 in untreated and IR-treated conditions. However, the physical association of RNF8 with the p97EQ variant was increased ~5-fold when compared to p97-WT, revealing it to be a trapped p97 substrate.

Overall, these data suggest that the p97 ATPase forms a stable physical complex with the E3 ubiquitin ligase RNF8 under physiological and genotoxic conditions *in vivo* to prevent RNF8 hyper-accumulation.

### Homeostasis of RNF8 is controlled by auto-ubiquitination and the ubiquitin–proteasome system

RNF8 is an E3 ubiquitin ligase that, in association with E2-conjugating enzymes Ubc13, Ubc8 and Ube2S, forms K63-Ub, K48-Ub or K11-Ub chains, respectively, on various substrates (Feng & Chen, 2012; Lok *et al*, 2012; Mallette *et al*, 2012; Zhang *et al*, 2013; Thorslund *et al*, 2015; Paul & Wang, 2017). As the K48-Ub is the main

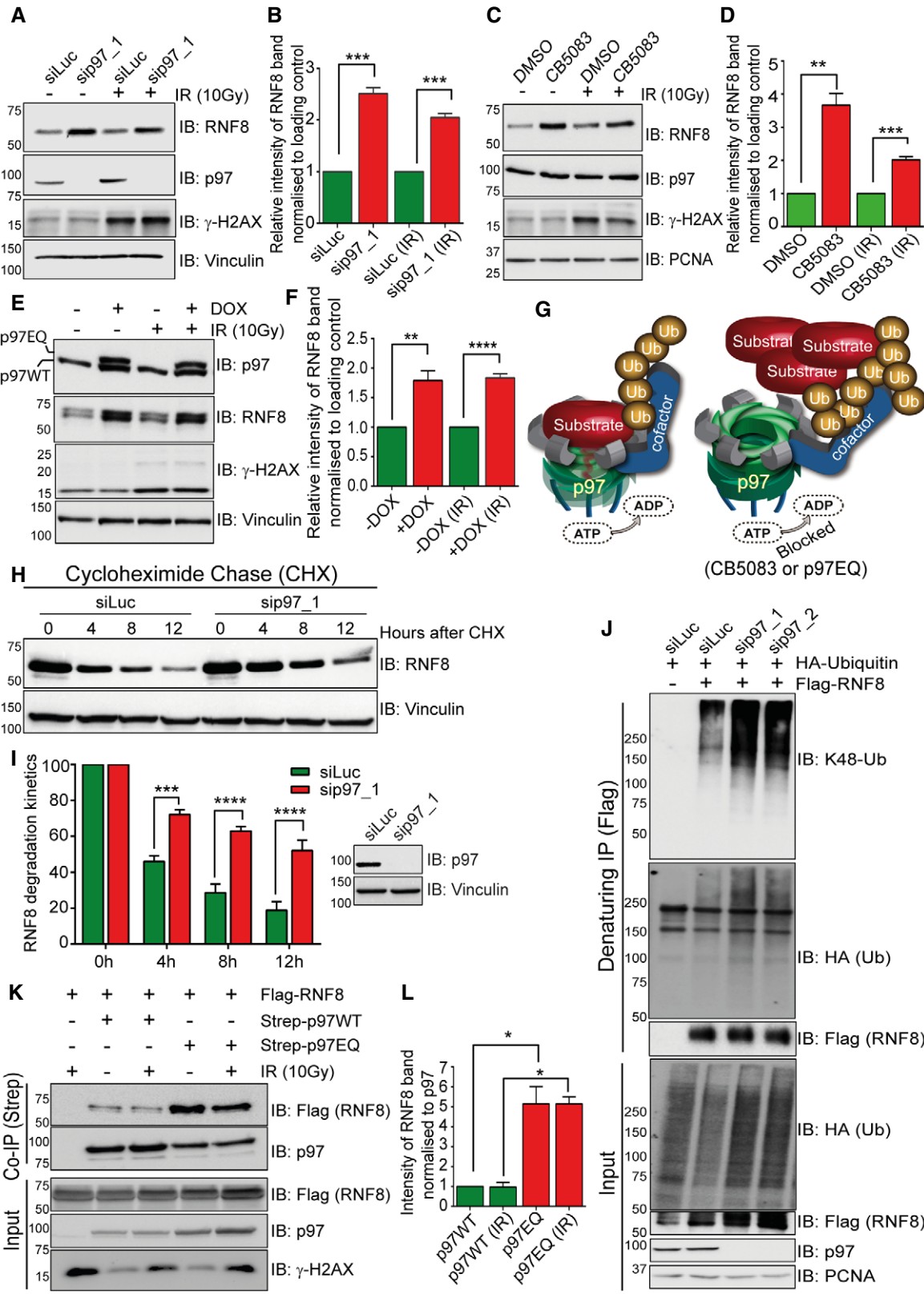

**Figure 1.**

◄

**Figure 1.  RNF8 is a ubiquitinated substrate of p97 under physiological and genotoxic conditions.**

A   Western blot analysis showing increased RNF8 protein level in HeLa cells after siRNA-mediated p97 depletion under physiological conditions and after IR (10 Gy).
B   Graph represents the quantifications of (A) (***$P < 0.001$; unpaired $t$-test, $n = 3$, mean + SEM).
C   Western blot analysis showing increased RNF8 protein level in HeLa cells after p97 chemical inhibition (CB5083, 10 μM for 6 h) under physiological conditions and after IR (10 Gy).
D   Graph represents the quantifications of (C) (**$P < 0.01$, ***$P < 0.001$; unpaired $t$-test, $n = 3$, mean + SEM).
E   Western blot analysis showing increased RNF8 protein level in HEK293 cells after doxycycline-inducible mild expression of the p97EQ variant under physiological conditions and after IR (10 Gy).
F   Graph represents the quantifications of (E) (**$P < 0.01$, ****$P < 0.0001$; unpaired $t$-test, $n = 4$, mean + SEM).
G   Model representing the processing of ubiquitinated substrate by p97 ATPase activity. Inactivation of p97 ATPase activity leads to the accumulation of ubiquitinated substrate.
H   Western blot analysis of CHX chase kinetics showing reduced RNF8 degradation rate in HEK293 cells after siRNA-mediated p97 depletion.
I   Graph represents the quantifications of (H) (***$P < 0.001$, ****$P < 0.0001$; two-way ANOVA, $n = 3$, mean + SEM) and Western blot for efficacy of siRNA depletion of p97 (right).
J   Western blot analysis of Flag-RNF8 denaturing-IP in HEK293 cells showing K48-linked hyper-ubiquitination of RNF8 after siRNA-mediated p97 depletion.
K   Western blot analysis of Strep-p97 Co-IP in HEK293 cells showing increased RNF8 interaction with p97EQ variant as compared to p97-WT under physiological conditions and after IR (10 Gy).
L   Graph represents the quantifications of (K) (*$P < 0.05$; unpaired $t$-test, $n = 2$, mean + SEM).

Source data are available online for this figure.

signal for proteasomal degradation, we asked whether RNF8 is also able to ubiquitinate itself (auto-ubiquitination) with K48-Ub and whether this serves as a signal for its own degradation. Flag-RNF8-WT or Flag-RNF8 bearing one inactivating amino acid mutation (C403S) in its E3-ligase active centre (RING finger mutated variant, RING*) (Mailand *et al*, 2007) was expressed and isolated under denaturing conditions. While RNF8-WT was heavily K48-polyubiquitinated and to a much lesser extent K63-polyubiquitinated, the inactive RNF8-RING* variant was almost completely devoid of ubiquitination (Fig 2A). Additionally, siRNA-mediated depletion of p97 further enhanced polyubiquitination of RNF8-WT, but not RNF8-RING*, suggesting that RNF8 is K48-auto-ubiquitinated and subsequently processed by p97.

p97 is a central component in the ubiquitin-dependent proteasome degradation system, but recent data also suggest a role for p97 in the autophagy-dependent degradation of ubiquitinated substrates (Bug & Meyer, 2012). To investigate whether RNF8 homeostasis is under control of the ubiquitin–proteasome system, we monitored the rate of endogenous RNF8 degradation by cycloheximide (CHX) chase under physiological conditions (Fig 2B and C, and Appendix Fig S1A and B). The rate of RNF8 turnover/degradation was analysed in HeLa cells treated with either DMSO or the proteasome inhibitor MG132. Endogenous RNF8 degradation was strongly suppressed when the proteasome was inhibited (Fig 2B and C). Similar to p97-depleted cells (Fig 1J), inhibition of the proteasome strongly boosted the K48 ubiquitination of RNF8 under physiological conditions (Fig 2D). Additionally, the degradation kinetics of the Flag-RNF8-WT and Flag-RNF8-RING* variants were compared in U2OS cells where the endogenous RNF8 was depleted by RNF8 shRNA (Appendix Fig S1C). RNF8-WT was rapidly degraded, but the degradation of RNF8-RING* was completely blocked (Fig 2E and F). Altogether this suggests that RNF8 generates K48-Ub chains on itself and thus regulates its own turnover and homeostasis via p97 and the proteasome.

## Ataxin 3 is a nuclear p97-associated DUB that interacts with RNF8

The aforementioned results suggest that RNF8 is heavily auto-K48-polyubiqiutinated, triggering its degradation by the p97–proteasome system. This raises the question of how RNF8 is protected from premature/accelerated degradation. As p97 and the proteasome co-exist with several DUBs, we reasoned that a DUB associated with p97 and/or the proteasome might be involved in the regulation of RNF8 homeostasis.

To test this hypothesis, we performed a SILAC-based quantitative p97-interactome mass spectrometry. We focused on RNF8 in the nuclear compartment due to its crucial role in DSB repair. Thus, the p97-Strep-tag proteome was isolated as previously described (Ritz *et al*, 2011) from nuclear fractions of HEK293 cells before and after exposure to 10 Gy IR (Fig 3A, Table EV1). As expected, p97 interacted with its two main cofactors, the Npl4–Ufd1 complex and p47 and its associated cofactor UBXD7, inside the nucleus. Interestingly, the p97-associated DUB ATX3 was the only DUB identified in the nuclear p97 proteome, which mildly increased in association after IR. The p97 proteome also contained many proteasome subunits, but not the proteasome-associated DUBs Rpn11/PSMD14, Ubp6/USP14 or UCH37/UCHL5 (Collins & Goldberg, 2017). We therefore decided to focus on ATX3.

ATX3 is a member of the polyglutamine repeats (polyQ) disease family as it contains a polyQ tract at its C-terminal end (Fig 3B). It is a p97-associated DUB required for cytosolic and endoplasmic reticulum-associated degradation processes and is capable of cleaving K63-Ub, K48-Ub and mixed Ub chains *in vitro* (Zhong & Pittman, 2006; Winborn *et al*, 2008). ATX3 interacts with p97 via a VCP-binding motif (VBM), binds ubiquitin via three ubiquitin-interacting motifs (UIMs) and cleaves ubiquitin chains by a N-terminal enzymatic region with a catalytic cysteine at position 14 (C14) located in the N-terminal Josephin domain (Burnett *et al*, 2003). Interestingly, it was recently shown that p97 (CDC-48 in *Caenorhabditis elegans*) and ATX3 fine-tune the ubiquitin signal and regulate apoptotic response to DNA damage in *C. elegans* (Ackermann *et al*, 2016). However, the existence and role of the p97–ATX3 complex at sites of DNA lesions in vertebrates are not known. Biochemical fractionation of HEK293 cell extracts and immunofluorescence (IF) analysis of U2OS cells revealed a sub-population of ATX3 in the nucleus and on chromatin, further confirming our mass spectrometry data that ATX3 is also a nuclear DUB (Fig EV1F and G).

Co-immunoprecipitation experiments with either p97 or ATX3 validated the mass spectrometry findings demonstrating a physical interaction between p97 and ATX3 in the nucleus (Fig 3C and D). However, the mildly increased interaction between p97 and ATX3 after IR observed by the SILAC mass spectrometry was not very clear when analysed by Western blotting. Intriguingly, RNF8 and the core proteasome subunit alpha 6 (PSα6) were also identified within the immunoprecipitate under physiological conditions as

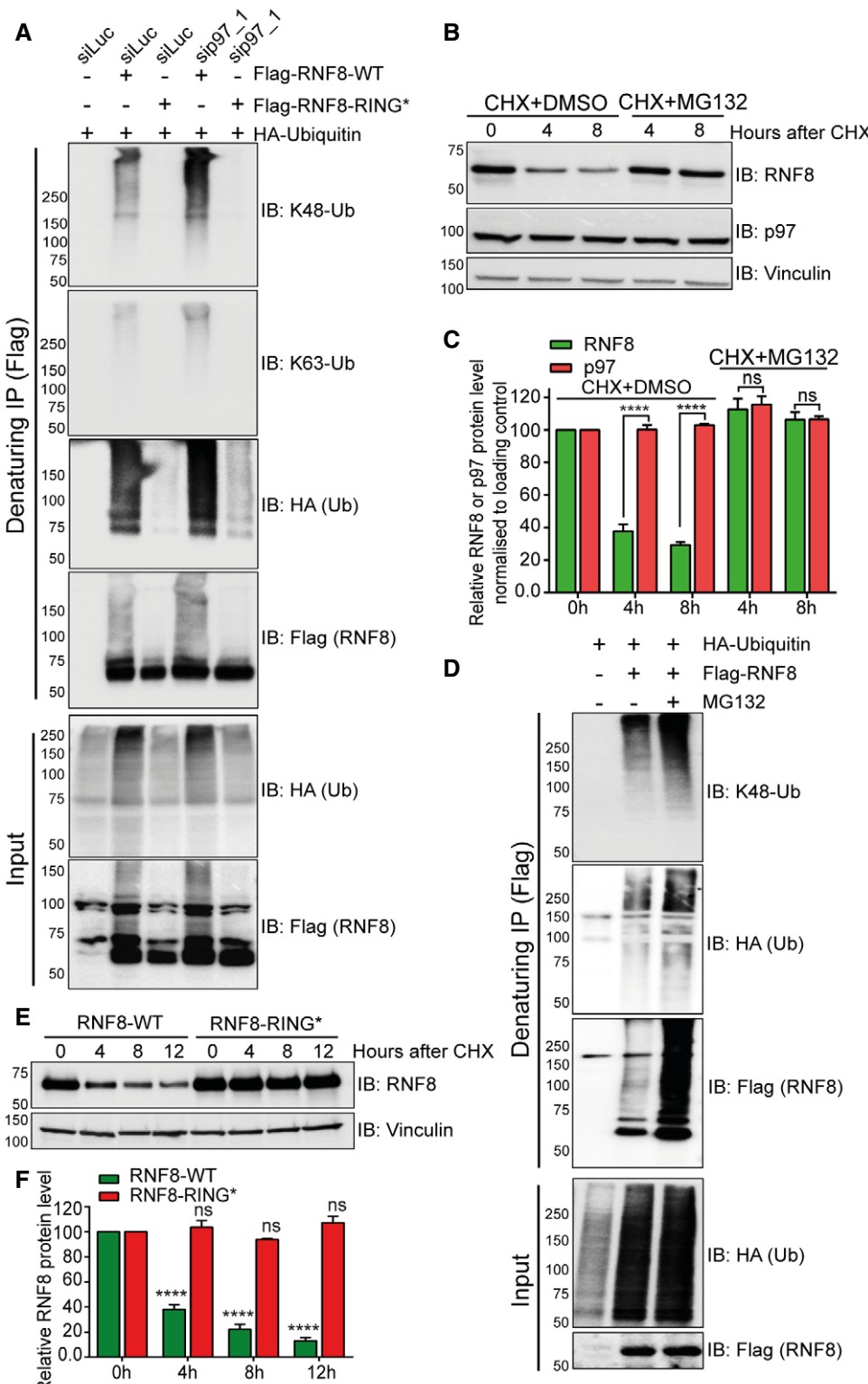

Figure 2.

◀

**Figure 2.  Homeostasis of RNF8 is controlled by auto-ubiquitination and the ubiquitin–proteasome system.**

A   Western blot analysis of Flag-RNF8 denaturing-IP in HEK293 cells showing the ubiquitination pattern of RNF8-WT and RNF8-RING* variant, under siRNA-mediated luciferase (siLuc) or p97-depleted conditions (sip97).

B   Western blot analysis of CHX chase kinetics in HeLa cells showing the degradation kinetics of RNF8 and inhibition of RNF8 degradation by simultaneous proteasome inhibition (MG132, 10 μM).

C   Graph represents the quantifications of (B) (ns: not significant, $P > 0.05$, ****$P < 0.0001$; two-way ANOVA, $n = 2$, mean + SEM).

D   Western blot analysis of Flag-RNF8 denaturing-IP in HEK293 cells showing hyper-ubiquitination of RNF8 after proteasome inhibition (MG132, 10 μM for 6 h).

E   Western blot analysis of CHX chase kinetics in U2OS cells, comparing the degradation rate of Flag-RNF8-WT and Flag-RNF8-RING*. Endogenous RNF8 was depleted by shRNF8 targeting only endogenous RNF8.

F   Graph represents the quantifications of (E) ($^{ns}P > 0.05$, ****$P < 0.0001$; two-way ANOVA, $n = 3$, mean + SEM).

Source data are available online for this figure.

well as post-irradiation (Fig 3D). This suggests the existence of a p97–ATX3–RNF8–proteasome complex *in vivo*. In order to test if p97 and ATX3 cooperate in the processing and homeostasis of RNF8, we overexpressed Flag-ATX3-WT or Flag-ATX3-VBM* (p97 interaction-deficient variant) and analysed the interaction of these two ATX3 variants with RNF8 (Fig 3E). ATX3-WT, but not ATX3-VBM*, formed a physical complex with p97. However, ATX3-VBM* retained the interaction with RNF8 and the main p97 cofactor complex Ufd1–Npl4, indicating that p97 cofactors, ATX3 (substrate-processing cofactor) and Ufd1–Npl4 complex (substrate-recruiting cofactors) can interact with RNF8 (substrate) independently of p97 (Fig 3E). Interestingly, p97 also binds RNF8 independently of ATX3, but the interaction between p97 and RNF8 was increased in ATX3-knockout (ΔATX3) cells (Fig 3F and G). This indicates that ATX3 can counteract the interaction between p97 and RNF8 and that most probably Ufd1–Npl4 bridges p97 with RNF8, as is the case for many well-characterised p97 substrates (Buchberger *et al*, 2015). Overall, these results suggest that RNF8 forms a complex with the p97 system composed of p97, Npl4–Ufd1 and ATX3 and that p97 and ATX3 bind RNF8 independently (Fig 3H).

ATX3 DUB activity is strongly reduced in the absence of p97 *in vitro* (Doss-Pepe *et al*, 2003; Laco *et al*, 2012). Likewise, we observed that the stability of ATX3 at the protein level strictly depends on p97 *in vivo*. Depletion of p97 destabilised ATX3 in HEK293 cells (Appendix Fig S1D). A similar phenotype was observed when HEK293 or HeLa cells were treated with the allosteric p97 inhibitor NMS873, which affects p97 conformation and thus prevents ATP hydrolysis (Magnaghi *et al*, 2013; Chou *et al*, 2014), but not with the p97 ATPase inhibitor DBeQ (Chou *et al*, 2011) (Appendix Fig S1E and F). We conclude that ATX3 can be only functional in the context of the p97 system.

### ATX3 deubiquitinates RNF8 to protect it from premature degradation

Since we confirmed an interaction between ATX3 and RNF8 *in vivo* (Fig 3), we analysed whether ATX3 regulates RNF8 homeostasis. First, we observed that the RNF8 protein level was lower for about 20% in ATX3-knockout cells when compared to wild-type cells (Fig 4A and B). Second, the rate of RNF8 degradation was monitored in ATX3-knockout or siRNA-depleted cells by CHX chase experiments. In both conditions, RNF8 was rapidly degraded (Figs 4A and B, and EV2A and B), and this was fully suppressed by proteasome inhibition (MG132) (Fig 4C). Importantly, ATX3 inactivation did not affect RNF8 transcription (Fig EV2C). This strongly

supports the idea that ATX3 is the DUB that counteracts RNF8 auto-ubiquitination and thus p97-facilitated degradation.

To further investigate how ATX3 affects RNF8 ubiquitination, we immunopurified Flag-RNF8 under denaturing conditions from cell extracts where either p97 or ATX3 was depleted (Fig 4D). Depletion of either p97 or ATX3 strongly enhanced RNF8 polyubiquitination, suggesting that the processing of ubiquitinated RNF8 is modulated by both p97 and ATX3. To directly prove that ATX3 deubiquitinates RNF8, we co-expressed Flag-RNF8 in either HeLa-WT or ATX3-knockout cells together with GFP-ATX3-WT, catalytically inactive GFP-ATX3-C14A or ubiquitin-binding-defective interacting motifs (UIMs*) GFP-ATX3-UIM* variants. RNF8 was immunoprecipitated under denaturing conditions, and the polyubiquitination pattern on RNF8 was analysed (Fig 4E). RNF8 displayed increased polyubiquitination in ATX3-knockout cells in comparison with WT cells (Fig 4E). These ubiquitin chains were mostly removed when ATX3-WT but not ATX3-C14A or ATX3-UIM* was expressed (Fig 4E; compare lanes 5, 6 and 7). The linkage type of the polyubiquitination was confirmed with K48-Ub chain-specific antibody.

In line with the above findings, accelerated RNF8 degradation was observed in ATX3-C14A- or ATX3-UIM*-expressing cells as compared to ATX3-WT (Fig 5A–D). This accelerated degradation of RNF8 was rescued significantly by reintroducing ATX3-WT but not the ATX3-VBM variant (Fig 5E and F). Interestingly, inactivation of either ATX3 or RNF8 without any genotoxic stress caused increased level of micronuclei formation (Fig 5G and H).

Altogether, these data show that ATX3, via its deubiquitinating activity (C14) and ubiquitin-binding capacity (UIMs), regulates RNF8 stability by removing its associated K48-polyubiquitin chains from (i) RNF8 in close association with p97 (VBM), and (ii) both proteins, ATX3 and RNF8, are essential for maintenance of genomic stability under physiological conditions.

### The p97–ATX3–RNF8 axis is essential for cell survival after IR-induced genotoxic stress

Our findings that p97 and ATX3 regulate RNF8 homeostasis under physiological conditions led us to speculate that this could be of importance when DNA damage occurs in cells as RNF8 is critical for accurate DSB repair and cell survival.

We performed neutral DNA comet assays to directly quantify amounts of genomic DSBs (Fig 6A; Nickson *et al*, 2017; Carter *et al*, 2018). Depletion of ATX3 with two different siRNAs significantly increased the level of DSBs even in unchallenged conditions. This result together with the increased levels of micronuclei in

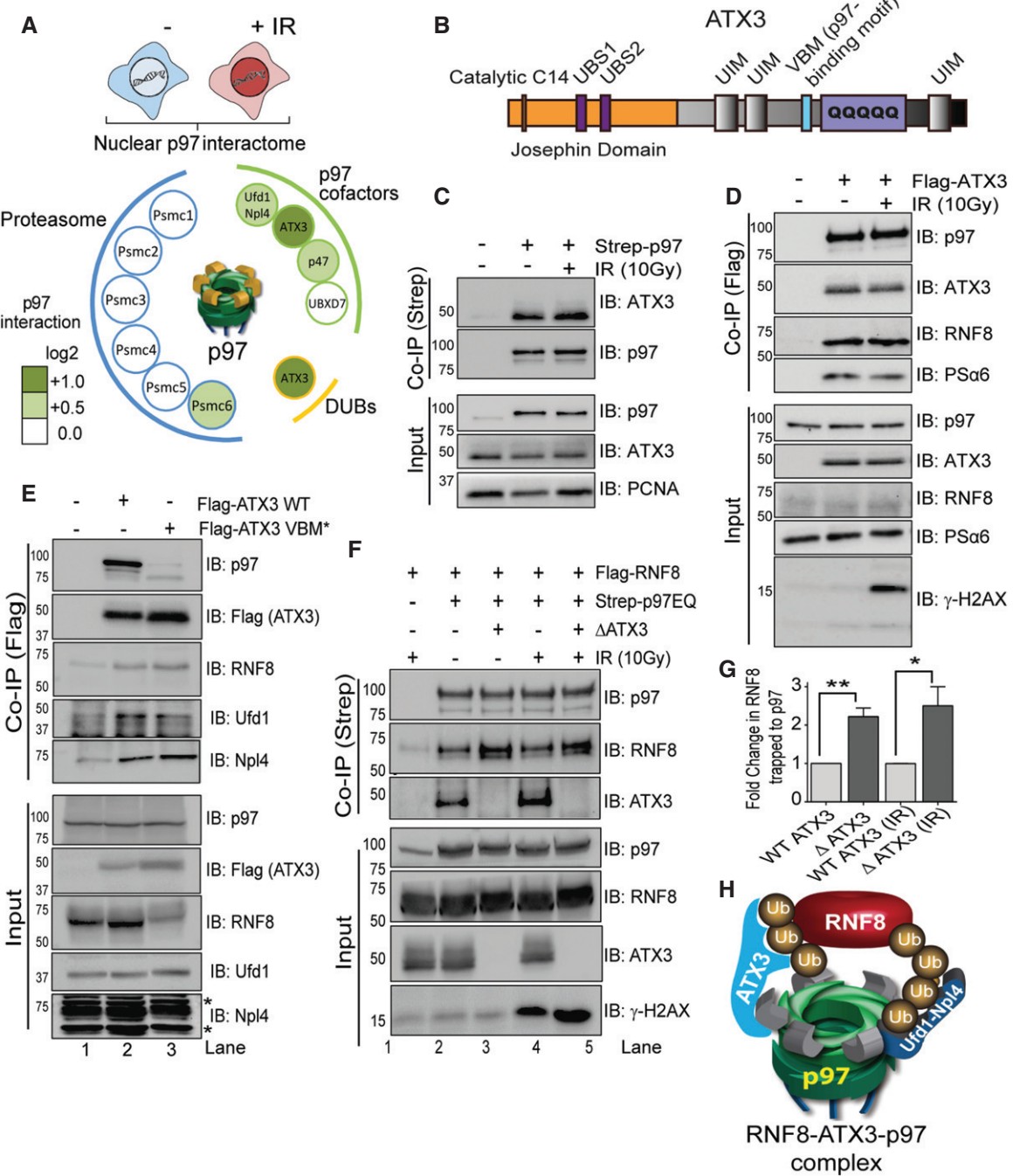

**Figure 3. Ataxin 3 is a nuclear p97 associated DUB that interacts with RNF8.**

A  Schematic representation of a SILAC-based mass spectrometry approach (upper graph). The p97 interactome showing ATX3 as the only DUB in nucleus with increased interaction with p97 after IR (10 Gy; lower graph).

B  Schematic representation of ATX3 domain structure. C14: cysteine catalytic centre at amino acid 14; UBS1 or 2: ubiquitin-binding site 1 or 2; UIM: ubiquitin-interacting motif; VBM: p97 binding motif; polyQ: polyglutamine region.

C  Western blot analysis of Strep-p97 Co-IP in nuclear fraction of HEK293 cells showing interaction of p97 with ATX3 under physiological conditions and after IR (10 Gy).

D  Western blot analysis of Flag-ATX3 Co-IP in HEK293 cells showing interaction of ATX3 with p97, RNF8 and PSα6 under physiological conditions and after IR (10 Gy).

E  Western blot analysis of Flag-ATX3 WT or Flag-ATX3 VBM* Co-IP in HEK293 cells showing interaction of ATX3 with p97, RNF8 and p97 core cofactors Npl4–Ufd1 (* represents unspecific bands).

F  Western blot analysis of Strep-p97EQ Co-IP in HEK293 cells, in the presence or absence (CRISPR knockout; ΔATX3) of ATX3, showing increased interaction of RNF8 with p97 in the absence of ATX3, under physiological conditions and after IR (10 Gy).

G  Graph represents quantifications of (F) (*$P < 0.05$, **$P < 0.01$; unpaired *t*-test, $n = 3$, mean + SEM).

H  Model showing the interaction of p97 and ATX3 with ubiquitinated RNF8.

Source data are available online for this figure.

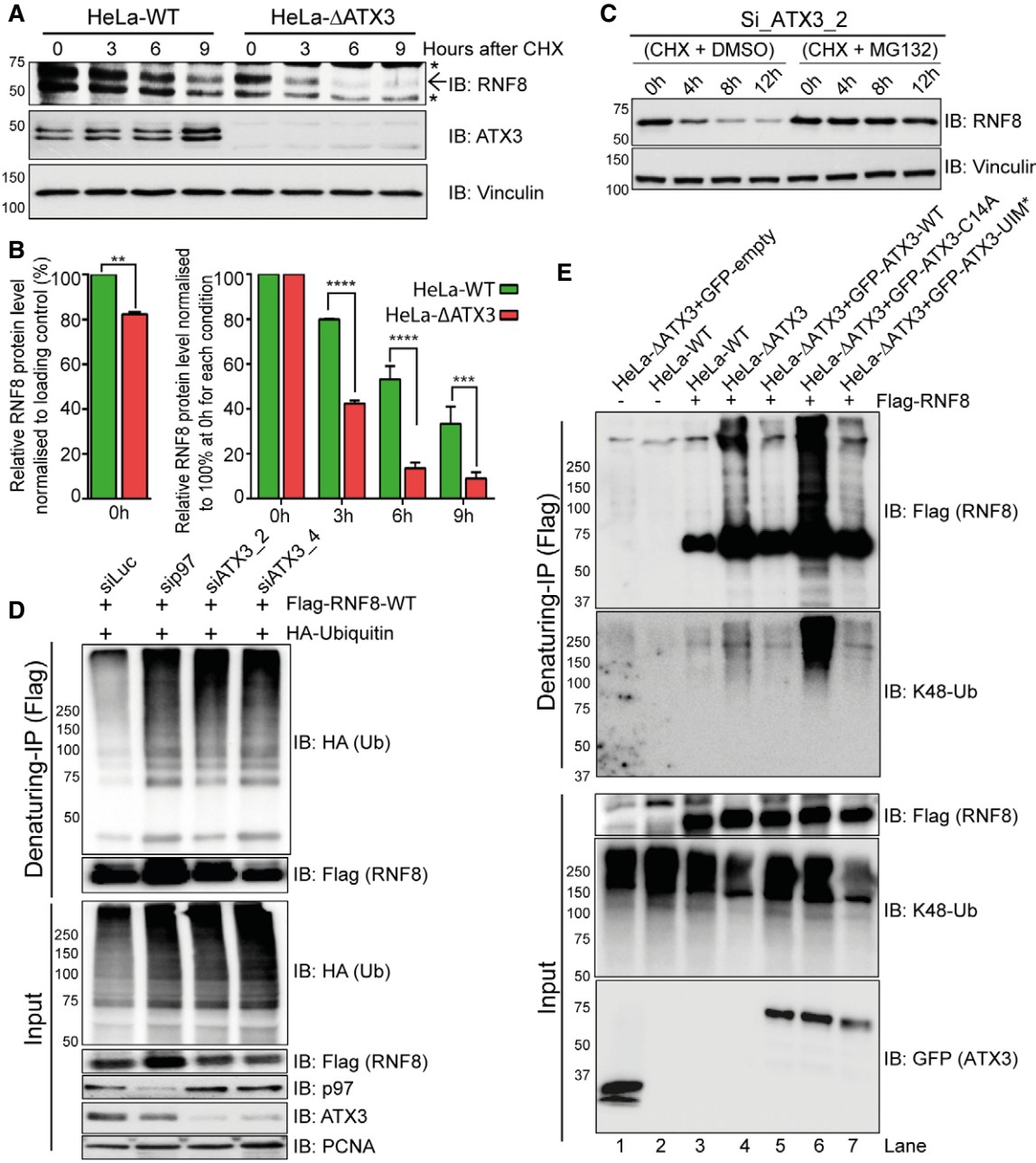

**Figure 4. ATX3 deubiquitinates RNF8.**

A   Western blot analysis of CHX chase kinetics in HeLa cells showing accelerated endogenous RNF8 degradation in the soluble fraction (cytosol and nucleosol) of ΔATX3 cell extract. Arrow represents the main RNF8 band, and asterisks represent unspecific bands.

B   Graphs represent the quantifications of (A). RNF8 level at starting point (0 h) was shown without equalisation (left). In order to nullify the difference in RNF8 level at starting point (0 h), we equalised RNF8 level to 100% and then compared the degradation rate (right) (**$P < 0.01$, ***$P < 0.001$, ****$P < 0.0001$; two-way ANOVA, $n = 2$, mean + SEM).

C   Western blot analysis of CHX chase showing the kinetics of endogenous RNF8 degradation in the soluble fraction (cytosol and nucleosol) of ATX3-knockdown HeLa cells. The degradation was completely blocked after simultaneous inhibition of proteasome (MG132, 10 μM).

D   Western blot analysis of Flag-RNF8 denaturing-IP in HEK293 cells showing RNF8 hyper-ubiquitination in siRNA-mediated p97 or ATX3-depleted conditions.

E   Western blot analysis of Flag-RNF8 denaturing-IP in HeLa cells showing RNF8 hyper-ubiquitination in ΔATX3 condition. RNF8 hyper-ubiquitination was suppressed by reintroduction of GFP-ATX3-WT but not with GFP-ATX3-C14A or GFP-ATX3-UIM*.

Source data are available online for this figure.

unchallenged ATX3- or RNF8-depleted U2OS cells further demonstrated the essential role of ATX3 and RNF8 in ensuring genome stability under physiological conditions. Moreover, the repair of IR-induced DSBs was also strongly impeded in ATX3-inactivated cells (Fig 6A).

We next monitored cell survival after IR treatment in either p97- or ATX3-depleted HeLa cells. As previously shown (Meerang et al, 2011), depletion of p97 hyper-sensitised cells to IR. ATX3-depleted cells were also hypersensitive to IR, to a similar extent as p97-depleted cells (Fig 6B). The role of ATX3 in protecting cells from IR was further confirmed in ATX3-knockout cells (Fig 6C). Importantly, the cellular hypersensitivity to IR in the absence of ATX3 was rescued to the level of control cells when ATX3-WT but not

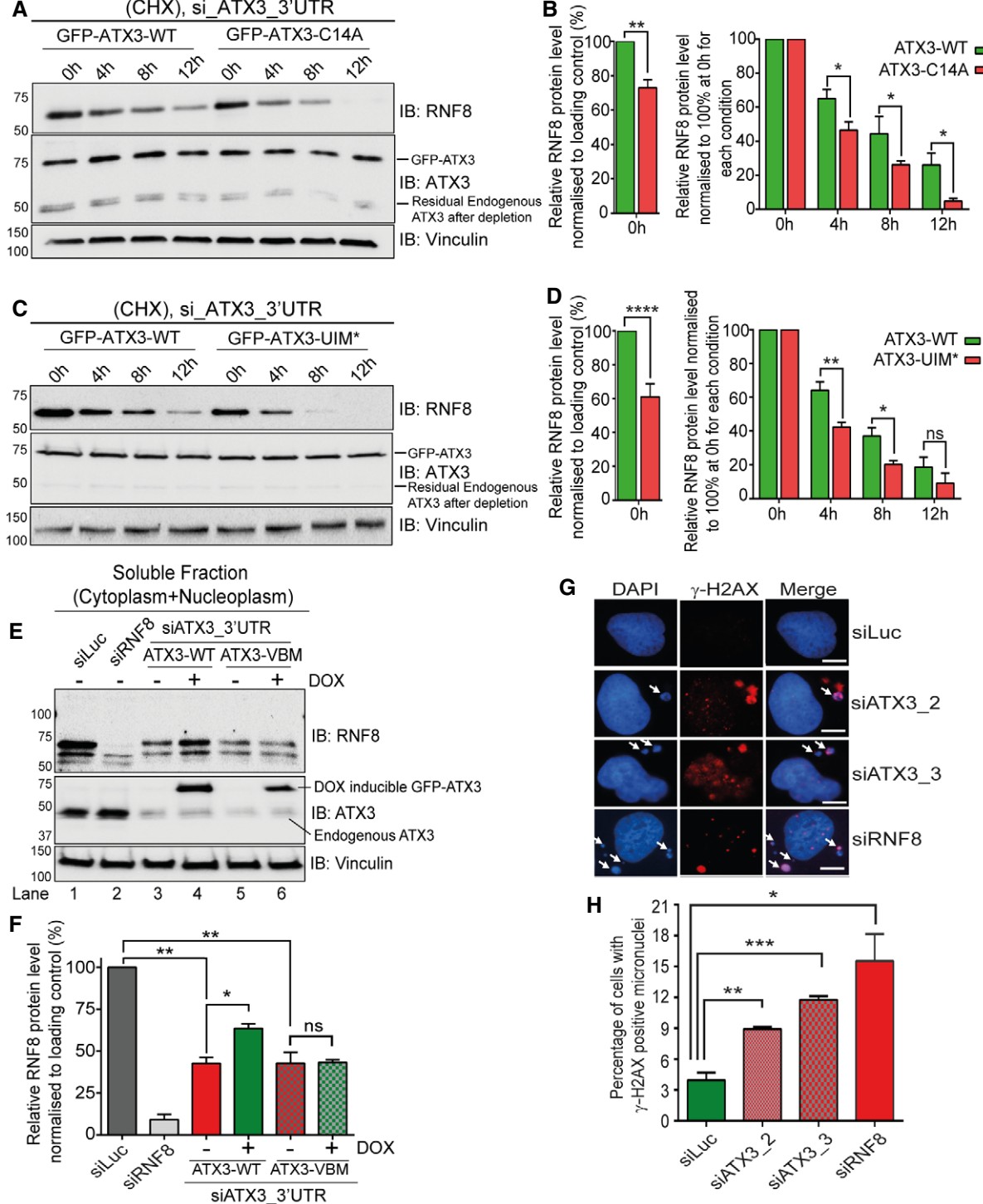

**Figure 5.**

**Figure 5. ATX3 protects RNF8 from premature degradation.**

A  Western blot analysis of CHX chase showing accelerated degradation of RNF8 in soluble fraction (cytoplasm + nucleoplasm) of HeLa cells, expressing DOX-inducible GFP-ATX3-C14A variant as compared to GFP-ATX3-WT. Endogenous ATX3 was depleted with siRNA targeting 3′UTR region of ATXN3.

B  Graph represents the quantifications of (A). In order to nullify the difference in RNF8 level at starting point (0 h), we equalised RNF8 level to 100% and then compared the degradation rate. RNF8 level at starting point (0 h) was also shown without equalisation (*$P < 0.05$, **$P < 0.01$; two-way ANOVA, $n = 2$, mean + SEM).

C  Western blot analysis of CHX chase showing accelerated degradation of RNF8 in soluble fraction (cytoplasm + nucleoplasm) of HeLa cells, expressing DOX-inducible ATX3-UIM* variant as compared to ATX3-WT. Endogenous ATX3 was depleted with siRNA ATX3_3′UTR.

D  Graph represents the quantifications of (C). In order to nullify the difference in RNF8 level at starting point (0 h), we equalised RNF8 level to 100% and then compared the degradation rate. RNF8 level at starting point (0 h) was also shown without equalisation ($^{ns}P > 0.05$, *$P < 0.05$, **$P < 0.01$, ****$P < 0.0001$; two-way ANOVA, $n = 2$, mean + SEM).

E  Western blot analysis showing endogenous RNF8 protein level in soluble fraction (cytosol and nucleosol) of HeLa cells under indicated conditions. RNF8 level was significantly reduced after siRNA-mediated ATX3 depletion and then was significantly rescued by DOX-inducible expression of GFP-ATX3-WT but not with GFP-ATX3-VBM.

F  Graphs represent the quantifications of (E) ($^{ns}P > 0.05$, *$P < 0.05$, **$P < 0.01$; unpaired $t$-test, $n = 3$, mean + SEM).

G  Representative IF images showing $\gamma$-H2AX-positive micronuclei (white arrows) under physiological conditions with indicated siRNA-mediated depletion. Scale bar: 10 μm.

H  Graph represents quantification of (G), showing percentage of cells with $\gamma$-H2AX-positive micronuclei in U2OS cells (*$P < 0.05$, **$P < 0.01$, ***$P < 0.001$; unpaired $t$-test, $n = 3$, mean + SEM, at least 200 randomly selected nuclei were counted per condition per experiment).

Source data are available online for this figure.

ATX3-C14A, ATX3-UIM* (Fig 6D) or ATX3-VBM (Fig 6E) variants were expressed. This suggests that enzymatic processing of ubiquitinated substrates by ATX3 is critical for DSB repair and cell survival after IR treatment and that this role of ATX3 is strictly p97-dependent.

## The p97–ATX3 complex suppresses overaccumulation of RNF8 and K63-Ub at sites of DNA damage

To elucidate mechanistic insight of the p97–ATX3–RNF8 complex in DSB repair, we focused on sites of DNA lesions induced by either IR or UV-laser micro-irradiation. Both approaches cause DSB formation where RNF8-dependent ubiquitination is vital for orchestration of the DDR and consequently repair and cell survival (Kolas *et al*, 2007; Mailand *et al*, 2007; Feng & Chen, 2012).

As p97 and ATX3 also relocate quickly to sites of DNA damage (Fig EV2D), we asked whether the p97–ATX3 complex regulates RNF8 association with DSB sites. We therefore monitored RNF8 levels, and the ubiquitin signal as a read-out of RNF8 activity, after DNA damage induction in U2OS cells. Similar to the findings with the soluble pool of RNF8, depletion (Appendix Fig S1G) or chemical inhibition (CB5083) of p97 caused hyper-accumulation of RNF8 (Figs 6F and G, and EV2E–G) and K63-Ub (Fig 6I and J) at the sites of DNA damage at both time points analysed. Surprisingly, depletion of ATX3 with three different siRNAs also caused RNF8 and K63-Ub hyper-accumulation although to a lesser extent than in p97-depleted cells. The specificity of the antibody for endogenous RNF8 was confirmed by siRNA-mediated RNF8 depletion (Appendix Fig S1H).

While this manuscript was in preparation, it was reported that siRNA-mediated ATX3 depletion attenuates RNF8 recruitment and K63-Ub formation at sites of DNA damage (Pfeiffer *et al*, 2017). To address this, we have investigated RNF8 protein levels at the sites of DNA lesions by using three different sources of DNA damage: IR (Fig 6F–H), UV-laser micro-irradiation (Fig EV2E–G) and two-photon laser micro-irradiation (Fig EV2K and L), and two different antibodies against endogenous RNF8 (Fig EV2E–G and H); and we also analysed the behaviour of constitutively expressed GFP-RNF8 in fixed or living cells (Figs 6F and G, and EV2K and L). Our cumulative observations all support that RNF8 protein accumulates at

sites of DNA damage under ATX3-deficient conditions. We also analysed the recruitment of endogenous and ectopically expressed Flag-RNF8 at the sites of UV-induced DNA damage in ATX3-depleted U2OS cells (Fig EV2E–G, I and J) as well as in ATX3-knockout HeLa cells (Appendix Fig S1I and J). Both RNF8 versions, endogenous and ectopically expressed, also hyper-accumulated at the sites of DNA damage in ATX3-knockout HeLa cells. In addition, RNF8 also increased its accumulation on total chromatin fraction in ATX3-inactivated cells and was modified, most probably by ubiquitination (Appendix Fig S1K and L). Interestingly, inactivation of the known p97 cofactor complex for DSB repair, the Ufd1–Npl4 heterodimer (Meerang *et al*, 2011; van den Boom *et al*, 2016), by siRNA-mediated Npl4 depletion did not cause RNF8 hyper-accumulation at sites of IR-induced DNA lesion in fixed cells (Fig EV2M and N). Furthermore, the inactivation of Ufd1-Npl4 complex, by either siRNA-mediated Npl4 or Ufd1 depletion, also did not cause any RNF8 hyper-accumulation at the sites of two-photon micro-irradiation damage in living cells (Fig EV2K). However, inactivation of p97 by chemical inhibition or siRNA-mediated ATX3 depletion again strongly induced RNF8 accumulation at the sites of IR or two-photon micro-irradiation DNA lesions in fixed or living cells, respectively (Fig EV2K and L). This suggests the existence of a p97–ATX3 complex, which, independently of the known Ufd1-Npl4 core cofactor complex for DSB repair, extracts RNF8 from the sites of DNA damage.

These results also raised the question of whether MDC1, as the platform for RNF8 recruitment to sites of DSB, is one of the most relevant ATX3 substrates in DSB repair as reported (Pfeiffer *et al*, 2017). Thus, we tested whether MDC1 is present in the ATX3 complex under physiological or IR-induced conditions (Fig EV3A). Despite endogenous MDC1 being strongly visible in the input fraction, MDC1 signal was not detectable in GFP-ATX3 co-immunoprecipitate under our experimental conditions. In contrast, endogenous RNF8 or even KU70 co-precipitated with ectopically expressed ATX3 (Figs EV3A, and 3D and E).

Analysis of IR-induced MDC1 foci formation and resolution at the sites of DNA damage ($\gamma$-H2AX) (Fig EV3B and C) and MDC1 phosphorylation status (Fig EV3D) indicated that ATX3-knockout cells have a functional MDC1 response after IR. Moreover, ATX3-knockout

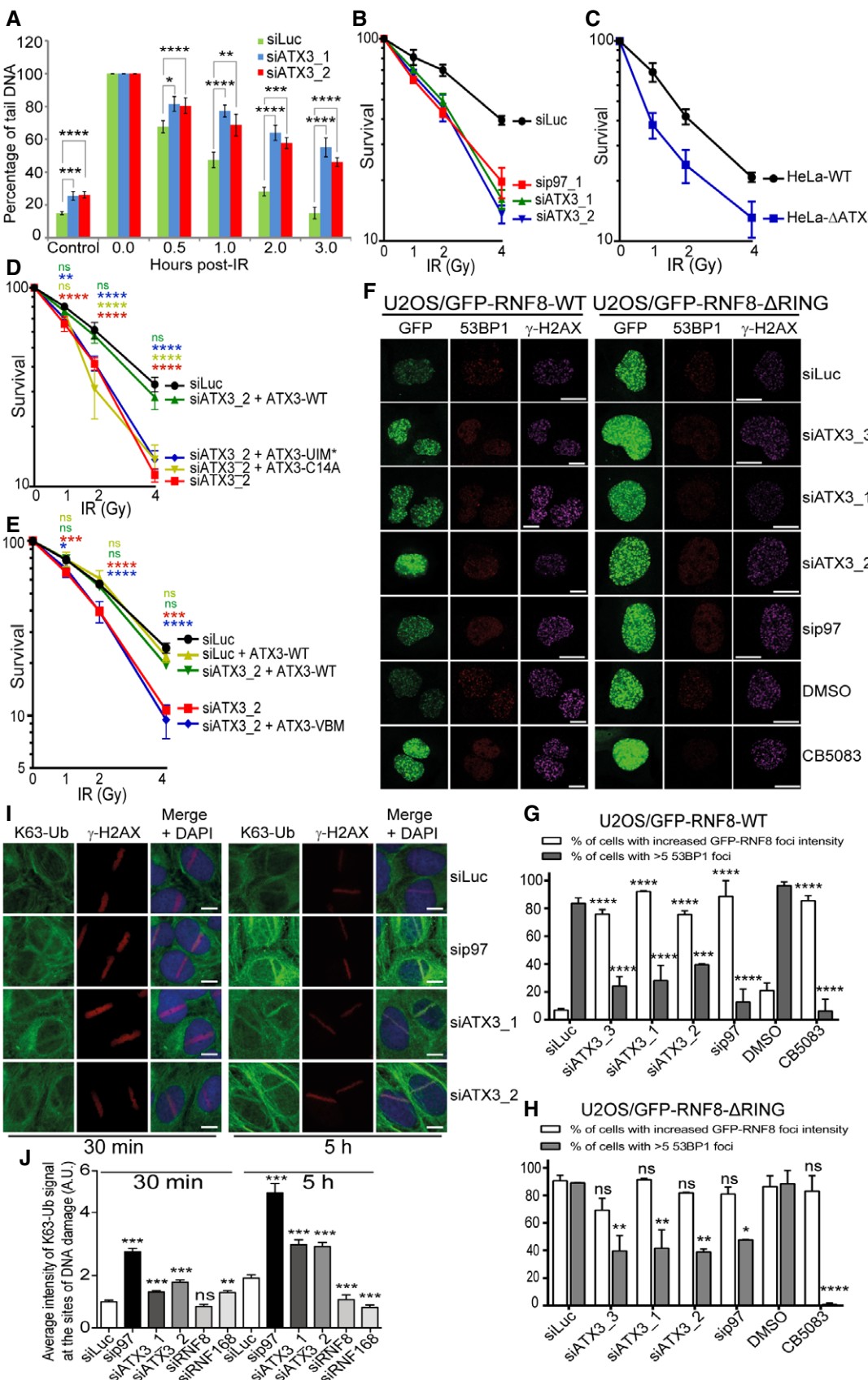

Figure 6.

◄

**Figure 6. The p97–ATX3 complex suppresses hyper-accumulation of RNF8 and K63-Ub at sites of DNA damage.**

A   Quantifications of neutral comet assay showing the level of DSBs under indicated siRNA-depleted conditions, before and after IR (4 Gy) (*$P < 0.05$, **$P < 0.01$, ***$P < 0.001$, ****$P < 0.0001$; two-way ANOVA, $n = 3$, mean + SEM).

B   Colony formation assay (CFA) showing the IR sensitivity of HeLa cells under siRNA-mediated p97 or ATX3-depleted conditions ($n = 3$, mean ± SD).

C   CFA showing the IR sensitivity of HeLa-WT cells compared to HeLaΔATX3 cells ($n = 3$, mean ± SD).

D   CFA showing the rescue of cell survival in IR-treated, ATX3-depleted (siRNA), HeLa, cells, after DOX-inducible expression of ATX3-WT but not with ATX3-C14A or ATX3-UIM* variant (ns$P > 0.05$, **$P < 0.01$, ****$P < 0.0001$; two-way ANOVA, $n = 3$, mean ± SD).

E   CFA showing the rescue of cell survival in IR-treated, ATX3-depleted (siRNA), HeLa, cells, after DOX-inducible expression of ATX3-WT but not with ATX3-VBM variant (ns$P > 0.05$, *$P < 0.05$, ***$P < 0.001$, ****$P < 0.0001$; two-way ANOVA, $n = 3$, mean ± SD).

F   Representative IF micrographs in U2OS cells showing IRIF formation of GFP-RNF8-WT, 53BP1 and γ-H2AX (first panel) and GFP-RNF8-ΔRING, 53BP1 and γ-H2AX (second panel) under indicated siRNA-depleted or p97-inhibited (CB5083) conditions. Scale bar: 10 μm.

G   Quantification of (F) for GFP-RNF8-WT. Graph represents the percentage of cells with increased GFP-RNF8-WT foci (white bars) or percentage of cells with > 5 53BP1 foci (grey bars) (***$P < 0.001$, ****$P < 0.0001$; two-way ANOVA, $n = 2$, mean + SEM, at least 100 nuclei per condition and experiment).

H   Quantification of (F) for GFP-RNF8-ΔRING. Graph represents the percentage of cells with increased GFP-RNF8-ΔRING foci (white bars) or percentage of cells with > 5 53BP1 foci (grey bars) (ns$P > 0.05$, *$P < 0.01$, **$P < 0.01$, ****$P < 0.0001$; two-way ANOVA, $n = 2$, mean + SEM, at least 100 nuclei per condition and experiment).

I   Representative IF images and quantification showing endogenous K63-Ub and γ-H2AX at UV-A micro-laser-induced DNA damage tracks in U2OS cells under indicated siRNA-depleted conditions. Scale bar: 10 μm.

J   Quantification of (I). Graph represents the average intensity of the K63-Ub (ns$P > 0.05$, **$P < 0.01$, ***$P < 0.001$; unpaired *t*-test, $n = 2$, mean + SEM, more than 50 nuclei were analysed).

cells displayed delayed clearance of both MDC1 and γ-H2AX from DSB sites, suggesting defective DSB repair.

Overall, we concluded that ATX3 inactivation causes retention and consequently increased accumulation of RNF8 at DSB sites. Increased accumulation of RNF8 in p97 or ATX3-inactivated cells was directly responsible for the increased accumulation of K63-Ub at damage sites, as co-depletion of RNF8 completely suppressed this phenotype (Fig EV3E and F). Additionally, RNF8 auto-ubiquitination was directly responsible for the removal of RNF8 from sites of DNA damage, as expression of RNF8-RING* caused strong retention and hyper-accumulation of RNF8-RING* at the sites of DNA damage at either time point (Fig EV3G and H). This finding was also confirmed at sites of DSBs induced by IR in U2OS cells (Fig 6F and H).

Short inhibition of proteasome activity with MG132 causes depletion of the nuclear ubiquitin pool (Dantuma *et al*, 2006). MG132 treatment led to the accumulation of RNF8 (Appendix Fig S1M and N), which further supports that the turnover of RNF8 from sites of DNA damage is a ubiquitin-dependent process. Strikingly, depletion of p97 increased the accumulation of both K48-Ub (Fig EV4A and B) and K63-Ub (Fig 6I and J) at sites of DNA damage (Baranes-Bachar *et al*, 2018), indicating that several different substrates of p97 are accumulating in its absence. However, inactivation of ATX3 only increased the accumulation of K63-Ub at sites of DNA damage. This corresponds to the increased amount of RNF8 at sites of DNA lesion and suggests that RNF8 remains enzymatically active. Therefore, the p97–ATX3 complex selectively extracts a subset of p97 substrates from DNA lesions.

Overall, our data demonstrate that the p97–ATX3 complex prevents excessive accumulation of RNF8 at sites of DNA damage, and this removal depends on RNF8 E3-ligase activity and the proteasome.

### The p97–ATX3 complex fine-tunes histone H1 ubiquitination after DSBs

In contrast to RNF8 accumulation at DSB sites, recruitment of the downstream factors RNF168 and 53BP1 was reduced at the early time point (30 min) post-UV-laser irradiation (Fig 7A–D) in both p97- or ATX3-depleted cells. Similar results were obtained for 53BP1

when DSBs were induced by IR in GFP-RNF8-expressing cells (Fig 6F and G). This suggests that unrestrained K63 ubiquitination at DSB sites due to p97–ATX3 inactivation may be leading to dysfunctional DSB repair.

The essential signals for RNF168 recruitment are the K63-ubiquitin chains on histone H1 generated by RNF8 (Thorslund *et al*, 2015). We therefore monitored the status of histone H1 ubiquitination in either p97- or ATX3-depleted cells by biochemical isolation of histone H1 under denaturing conditions. Surprisingly, similar to RNF8 knockdown and consistent with defective RNF168 recruitment, histone H1 ubiquitination was strongly impaired in both p97- and ATX3-depleted cells at the early 30-min time point after IR (Fig 7E). However, at the late time point (5 h), histone H1 ubiquitination was restored to the levels observed in control cells in both p97- and ATX3-depleted cells (Fig EV4C and D) correlating with the increased recruitment observed for RNF168 and 53BP1 to sites of DNA damage in ATX3-depleted cells at the 5-h time point (Fig 7A–D, 5-h time point). The recruitment of RNF168 and 53BP1 was not rescued in the case of p97 depletion, indicating pleiotropic effects of p97 at the site of DSBs.

Overall, this shows that the p97–ATX3 complex extracts RNF8 from sites of DNA damage to facilitate the timely ubiquitination of H1 and recruitment of downstream factors RNF168/53BP1.

### p97–ATX3-mediated RNF8 homeostasis at sites of DNA damage orchestrates DSB repair pathway choice

In contrast to the defective recruitment of RNF168 and 53BP1 (Fig 7A–D), the recruitment of BRCA1 to either UV-micro-irradiated lesions (Fig 7F and G) or IR-induced foci (Fig EV4E–G) was not affected in ATX3-depleted conditions. Similar results were obtained for the BRCA1 ubiquitin-binding partner RAP80 (Fig EV5A and B). Unlike ATX3 inactivation where only one branch of DSB repair pathway signalling was affected (RNF168/53BP1 for NHEJ), inactivation of p97 impaired the recruitment of the key factors in both DSB repair pathways: BRCA1 and Rad51 (HR pathway) and 53BP1 (NHEJ pathway) (Figs 7A–F, and EV4E–G, and EV5C and D). These results suggest that there are several p97 complexes or even subcomplexes working at the sites of DSB lesions, and inactivation of

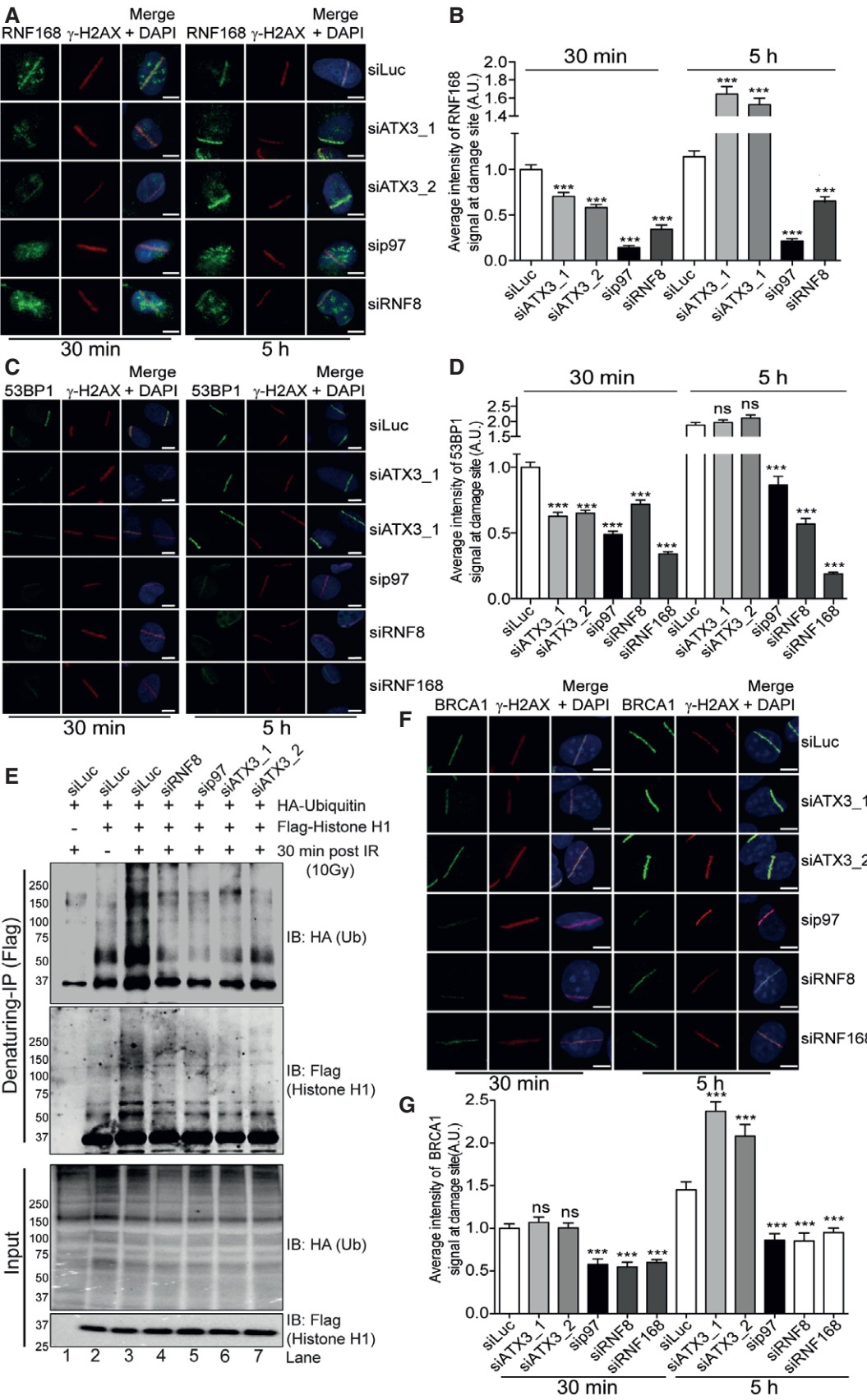

**Figure 7.**

Figure 7. The p97–ATX3 complex fine-tunes histone H1 ubiquitination after DSBs.

A   Representative IF images showing the average intensity of endogenous RNF168 at UV-A micro-laser-induced DNA damage tracks in U2OS cells. Scale bar: 10 μm.
B   Graph represents the quantification of (A) (***$P < 0.001$; unpaired $t$-test, $n = 3$, mean + SEM, more than 50 nuclei were analysed).
C   Representative IF images showing the average intensity of endogenous 53BP1 at UV-A micro-laser-induced DNA damage tracks in U2OS cells. Scale bar: 10 μm.
D   Graph represents the quantification of (C) ($^{ns}P > 0.05$, ***$P < 0.001$; unpaired $t$-test, $n = 2$, mean + SEM, at least 80 nuclei per condition and experiment).
E   Western blot analysis of Flag-histone H1 denaturing-IP, showing ubiquitination pattern of histone H1 after 30 min of IR (10 Gy) treatment under indicated siRNA-depleted conditions.
F   Representative IF images showing the average intensity of endogenous BRCA1 at UV-A micro-laser-induced DNA damage tracks in U2OS cells. Scale bar: 10 μm.
G   Graph represents the quantification of (F) ($^{ns}P > 0.05$, ***$P < 0.001$; unpaired $t$-test, $n = 2$, mean + SEM, at least 100 nuclei per condition and experiment).

Source data are available online for this figure.

p97 as their main conductor inactivates all of these complexes/sub-complexes and causes pleiotropic effects. The p97–ATX3 complex facilitates RNF8 turnover at DSB sites and thus positively and specifically influences the timely ubiquitination of histone H1 and recruitment of RNF168 and 53BP1 and, accordingly, promotes the NHEJ repair pathway.

To further investigate how ATX3 balances DSB pathway choice, we analysed the cell cycle profile in HEK293 and U2OS cells. ATX3-knockout or ATX3 siRNA-depleted cells caused a slight increase of cells in G1 phase and decreased cells in the S/G2 phase when compared to wild-type cells (Appendix Fig S2A–G). This is likely due to increased genomic instability in ATX3-inactivated cells. Despite this, the level of 5′-DNA end resection in S-phase positive cells, measured by either Rad51 or phospho-RPA foci formation, was significantly increased in ATX3-inactivated cells (Fig EV5E and F). This additionally suggests that ATX3 inactivation does not affect initial steps of the HR pathway, namely 5′-DNA end resection.

We concluded that the increased level of 5′-DNA end resection in ATX3-inactivated cells is a consequence of elevated amount of RNF8 at DSB sites and formation of the K63-Ub that is still able to recruit the RAP80-BRCA1 complex (Figs 7F, G, and EV4E–G, and EV5A, B). However, increased levels of enzymatically functional RNF8 at sites of DNA damage negatively influence timely ubiquitination of histone H1 and consequent recruitment of RNF168/53BP1 and execution of the NHEJ pathway.

To investigate this hypothesis, we used GFP-reporter assays to measure the efficiency of each repair pathway (Bennardo et al, 2008). Depletion of p97 markedly affected both DSB repair pathways (Fig 8A and B). However, depletion of ATX3 specifically affected the efficiency of NHEJ but not HR. To further validate these findings, we took advantage of isogenic human cell lines where either 53BP1 (MCF7) or BRCA2 (DLD1) was knocked out (Fig 8C and D). Cells were exposed to IR in the presence or absence of independent siRNAs against ATX3, and cell survival was monitored by colony formation assay. Depletion of ATX3 did not further sensitise 53BP1-knockout (NHEJ-deficient) cells to IR (Fig 8C), suggesting that ATX3 and 53BP1 are epistatic. However, depletion of ATX3 in BRCA2-knockout (HR-deficient) cells further radio-sensitised these cells causing synthetic lethality (Fig 8D), indicating that ATX3 and BRCA2 function in two distinct pathways for cell survival after IR. Altogether, these results further support our findings that ATX3 selectively fine-tunes DSB repair by preventing extensive RNF8/K63-ubiquitin signalling, which balances DSB repair pathway choice.

Finally, if this model is correct, then removal of RNF8 in ATX3-depleted cells should prevent excessive 5′-DNA end resection and restore cell survival after IR. As expected, depletion of ATX3 caused

increased Rad51 recruitment to DSBs and hypersensitivity to IR (Fig 8E–G). Importantly, co-depletion of ATX3 and RNF8 suppressed both excessive Rad51 recruitment and hypersensitivity of cells to IR (Fig 8E–G). These data and previous results demonstrate that (i) ATX3 directly regulates RNF8, (ii) ATX3 counteracts excessive RNF8-dependent 5′-DNA end resection to support the RNF168/53BP1/NHEJ pathway and (iii) ATX3 and RNF8 are antagonistic in terms of cell survival to IR.

## Discussion

Here, we discovered a mechanism that regulates the homeostasis of the E3 ubiquitin ligase RNF8 under physiological and genotoxic conditions (Fig 8H). We demonstrated that the ubiquitin-dependent machinery, composed of the ATPase p97 unfoldase/segregase and the DUB ATX3, regulates RNF8 turnover in a proteasome-dependent manner. Our results reveal a mechanism to explain how the p97–ATX3–RNF8 complex fine-tunes DSB repair by preventing excessive RNF8-dependent K63-Ub modifications and 5′-DNA end resection and consequently promoting the NHEJ pathway, an essential pathway for cell survival after IR and thus cancer cell resistance to radiotherapy (Jeggo et al, 2011; Jeggo & Lobrich, 2015). As both p97 and ATX3 are druggable enzymes (Chou et al, 2014) and p97 inhibitors are in clinical trials for the treatment of several cancer types (Zhou et al, 2015), our findings pave the way for using p97 and ATX3 inhibitors as potential radiosensitisers clinically. In addition, as many cancers have dysregulated DNA repair pathways, inactivation of the p97–ATX3 complex might be a promising strategy to induce synthetic lethality in those cancers where the HR pathway is inactivated, such as BRCA2-deficient cancers. Indeed, inactivation of ATX3 caused synthetic lethality in BRCA2-knockout but not in 53BP1-knockout cells, further supporting our findings that the p97–ATX3 complex positively stimulates NHEJ (53BP1-dependent) by suppressing excessive 5′-DNA end resection, an initial step of the HR (BRCA-dependent) repair pathway.

### Regulation of RNF8 during physiological and genotoxic conditions

Our biochemical and cell biology data acquired in several human cell lines suggest two modes of RNF8 homeostasis and regulation (Fig 8H; Model).

Mode I: The p97–ATX3 sub-complex preserves a soluble pool of RNF8. Soluble RNF8 is constantly K48-auto-ubiquitinated by its own E3-ligase activity, and this signals for p97-dependent proteasomal

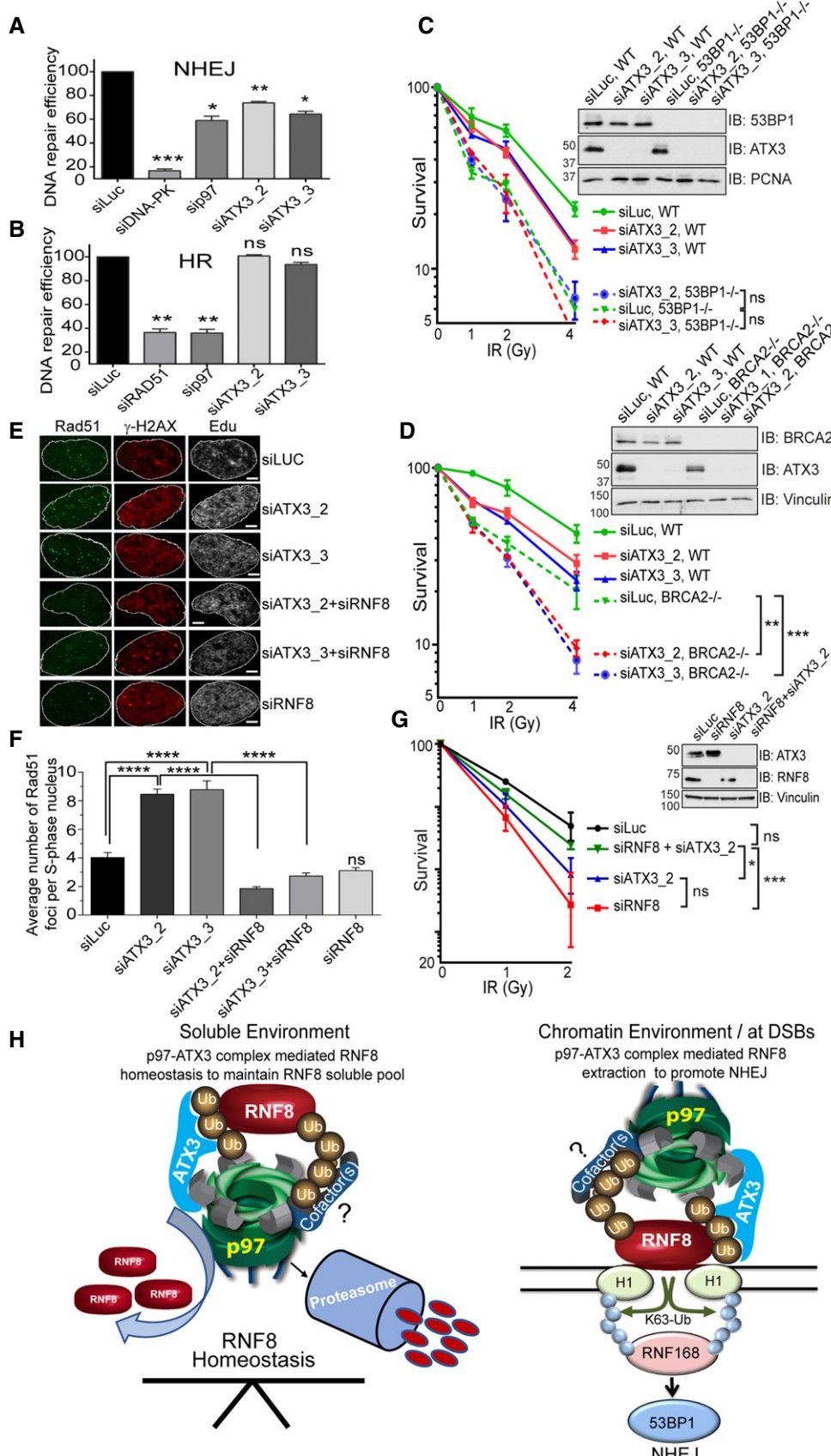

Figure 8.

Figure 8. The p97–ATX3 complex promotes accurate execution of NHEJ and DSB repair.

A  GFP based reporter assay showing the efficiency of NHEJ repair pathway under indicated siRNA-depleted conditions in HEK293 cells. Graph represents the quantifications from three independent experiments (*$P < 0.05$, **$P < 0.01$, ***$P < 0.001$; one-way ANOVA, $n = 3$, mean + SEM).

B  GFP based reporter assay showing the efficiency of HR repair pathway under indicated siRNA-depleted conditions in HEK293 cells. Graph represents the quantifications from three independent experiments ([ns]$P > 0.05$, **$P < 0.01$; one-way ANOVA, $n = 3$, mean + SEM).

C  CFA showing the IR sensitivity of 53BP1-proficient and 53BP1-deficient MCF7 cells under indicated siRNA-depleted conditions. Data were collected from two individual experiments with triplicates ([ns]$P > 0.05$; two-way ANOVA, $n = 2$, mean ± SEM).

D  CFA showing the IR sensitivity of BRCA2-proficient and BRCA2-deficient DLD1 cells under indicated siRNA-depleted conditions. Data were collected from two individual experiments with triplicates (**$P < 0.01$, ***$P < 0.001$; two-way ANOVA, $n = 2$, mean ± SEM).

E  Representative IF images showing Rad51 foci in S-phase (EdU-positive) U2OS nucleus after 5 h of IR (2 Gy) exposure. Scale bar: 10 μm.

F  Graph represents the quantification of (E) showing average number of Rad51 foci per S-phase (EdU-positive) nucleus ([ns]$P > 0.05$, ****$P < 0.0001$; unpaired $t$-test, $n = 2$, mean + SEM, at least 50 nuclei per condition and experiment).

G  CFA showing the rescue of cell survival in IR-treated, ATX3-depleted (siRNA), HeLa cells, after co-depletion (siRNA) of RNF8 ([ns]$P > 0.05$, *$P < 0.05$, ***$P < 0.001$; two-way ANOVA, $n = 3$, mean ± SEM).

H  Model of the p97–ATX3 complex in the regulation of RNF8 homeostasis under physiological condition (soluble environment) and in RNF8 chromatin extraction at sites of DSBs to promote NHEJ.

Source data are available online for this figure.

degradation. This pathway is counteracted by ATX3, which removes K48-Ub from RNF8 and thus slows down the rate of RNF8 degradation. Our results suggest that p97, ATX3 and RNF8 form a constitutive physical and functional complex, which is essential to maintain physiological expression levels of RNF8.

Mode II: In response to genotoxic stress, the p97–ATX3 sub-complex works synergistically to prevent increased accumulation of RNF8 at sites of DNA lesions by physical extraction of RNF8 from chromatin. High level of RNF8/K63-Ub at the sites of DNA lesions in p97–ATX3-inactivated cells correlates with delayed recruitment of downstream molecules including RNF168 and 53BP1 that are essential to facilitate NHEJ-mediated DSB repair. In this scenario, ATX3 promotes sufficient p97-dependent extraction of RNF8 in a classical ERAD-like manner (Wang *et al*, 2006; Zhong & Pittman, 2006). These findings are in line with the fact that p97 must associate with a DUB to process polyubiquitinated substrates (Bodnar & Rapoport, 2017a,b). The role of p97 in chromatin-associated degradation and consequently in DNA replication and repair was always linked to its main cofactor complex Ufd1–Npl4. In our experimental settings, inactivation of the Ufd1–Npl4 complex did not cause significant RNF8 hyper-accumulation at the sites of DNA lesion. This suggests that the p97–ATX3 complex is capable of promoting RNF8 chromatin extraction from the sites of DNA damage independently of its major cofactor involved in DSB repair, the Ufd1–Npl4 heterodimer. Altogether, this suggests the existence of distinct p97 complexes or even sub-complexes on chromatin and at sites of DNA damage, which process different substrates in a tightly controlled manner. This conclusion is additionally supported by results presented here that p97 inactivation causes increased accumulation of both K48-Ub and K63-Ub at the sites DNA lesions, yet ATX3 inactivation only affects K63-Ub. Further studies will be needed to address if additional cofactors are involved in the p97–ATX3 complex for processing of RNF8 at sites of DNA damage.

Importantly, our findings on the differential roles of ATX3 on chromatin-associated RNF8 (extraction) and soluble RNF8 (prevention of premature degradation) parallel previous studies on p97 and ATX3 in regulating the ER-associated T-cell receptors and p97–ATX3 substrates, CD3δ and TCRα (Zhong & Pittman, 2006). There, similar to our observation with RNF8, ATX3 was shown to regulate the level of ER soluble substrates (CD3δ and TCRα) by slowing down

their premature degradation, but at the same time, ATX3 assists p97 in CD3δ and TCRα extraction from the ER.

Taken together, our data suggest that ATX3 constantly regulates p97-dependent substrate processing, but the final fate of the substrates depends on their cellular localisation. When substrates are soluble, ATX3 prevents their premature degradation by removing K48-linked Ub chains to slow down the rate of p97-dependent degradation. When substrates are either embedded in the ER or bound to chromatin, then the p97 ATPase first needs to extract them from these compartments. Here, ATX3 assists the p97-dependent substrate extraction of the substrates from the ER or chromatin by trimming their associated ubiquitin chains to allow timely processing of the substrates (extraction from ER or chromatin), most probably through the central pore of p97 in a manner recently described (Bodnar & Rapoport, 2017b). The different roles of ATX3 on RNF8 could be potentially regulated by two different ubiquitin chain types on RNF8 (K48 and K63; Fig EV1D) and/or the phosphorylation status of ATX3 (Matos *et al*, 2016), as recently described for the DUB OTUD4 (Zhao *et al*, 2018). Unphosphorylated OTUD4 trims K48-Ub signals from alkylation DNA repair proteins and thus counteracts their proteasomal degradation. However, when phosphorylated, OTUD4 becomes a K63-Ub-specific protease that inhibits the K63-Ub-dependent inflammatory signalling pathway.

### The p97–ATX3 complex promotes NHEJ repair pathway

p97 is physically recruited to sites of DSBs, and its recruitment partially depends on DNA-PK phosphorylation or RNF8 and RNF168 ubiquitination (Livingstone *et al*, 2005; Acs *et al*, 2011; Meerang *et al*, 2011). This suggests that p97 is recruited to sites of DNA damage at multiple levels to remodel and extract various ubiquitinated substrates (Acs *et al*, 2011; Meerang *et al*, 2011; Ramadan, 2012; Puumalainen *et al*, 2014). p97 extracts the polycomb protein L3MBTL1, the KU70/80 heterodimer and Rad51, suggesting a multifaceted role of p97 in DSB repair (Acs *et al*, 2011; Bergink *et al*, 2013; van den Boom *et al*, 2016). It was recently demonstrated that p97 together with its major cofactor complex Ufd1–Npl4 strictly regulates the HR repair pathway by removing KU70/80 from DSBs to allow 5′-DNA end resection, Rad51 loading and HR repair activation (van den Boom *et al*, 2016). Moreover, van den Boom *et al*

demonstrated that chemical inhibition of p97 does not affect DSB repair after IR, which is in line with their model that p97 inactivation causes KU70/80 retention and presumably enhanced repair by the NHEJ pathway, a major pathway for IR-induced DNA damage repair (Jeggo *et al*, 2011). In addition, they suggested that p97 favours error-free HR over error-prone NHEJ. However, we here demonstrate that p97 inactivation affects both major DSB repair pathways (HR and NHEJ), and we identified a p97 complex, p97–ATX3, which stimulates the NHEJ repair pathway, by preventing excessive RNF8/K63-Ub accumulation that promotes 5′-DNA end resection and thus the HR pathway.

Even though it is often considered that RNF8 and RNF168 sit in the same signalling cascade, work from several laboratories demonstrated a clear separation between RNF8 and RNF168 function in DSB repair (Stewart *et al*, 2007; Hodge *et al*, 2016). For example, complementation of RNF8-knockout MEFs with RNF8 restores the recruitment of the downstream repair/signalling molecules (BRCA1 and 53BP1) important for both HR and NHEJ repair pathways. However, complementation of RNF8-knockout MEFs with RNF168 restores only recruitment of 53BP1 (NHEJ) but not BRCA1 (HR) to the sites of DNA lesions, therefore stimulating the NHEJ pathway (Hodge *et al*, 2016). Our results reported here and the recent discovery that RNF168-knockout mice have a functional HR pathway (Zong *et al*, 2019) further support the notion that the RNF8 signalling cascade is enough for functional HR.

We observed increased amounts of RNF8/K63-ubiquitin signal at sites of DNA damage in p97–ATX3-inactivated cells and attenuated ubiquitination of histone H1 (in the first 30 min after IR), accompanied by reduced RNF168 and 53BP1 recruitment. This raised the question of why histone H1 cannot be ubiquitinated even when RNF8 overaccumulates at damage sites (Thorslund *et al*, 2015). Importantly, RNF8, but not RNF168, has been shown to unfold higher-order chromatin structures (Luijsterburg *et al*, 2012). Thus, we speculate that a high accumulation of RNF8 in the presence of damage could make chromatin more accessible and affect histone H1 ubiquitination. Interestingly, H1 is evicted from sites of DNA damage (Sellou *et al*, 2016; Strickfaden *et al*, 2016), and recent high-throughput mapping of histone modifications at DSBs showed that breaks preferentially repaired by HR are enriched for ubiquitin but limited in histone H1 occupancy (Clouaire *et al*, 2018). These findings further support our observation that in ATX3-inactivated cells, the HR pathway is functional (enrichment of K63-Ub at sites of DSBs), but histone H1 is most probably not properly ubiquitinated as it is evicted from unfolded chromatin around HR-processed DSBs. However, further studies are required to address this important question in the context of the p97–ATX3–RNF8 complex and chromatin dynamics after DSB formation.

While our manuscript was in preparation, Pfeiffer *et al* (2017) reported that ATX3 is a DUB acting at DSB sites and removes ubiquitinated chains from MDC1 to prevent a premature removal of MDC1 from the breaks. Thus, in accordance with Pfeiffer *et al*, in ATX3-inactivated cells MDC1 is prematurely released from DNA damage sites. This reduces the recruitment of downstream molecules, such as RNF8 and RNF168 that are essential for both DSB repair pathways, HR and NHEJ. However, under our experimental conditions, we found that ATX3 forms a complex with RNF8 but not MDC1. Moreover, we also analysed the role of ATX3 in regulating

RNF8 association with DNA damage sites. In various human cell types, inactivation of ATX3 by either siRNA depletion or CRISPR/Cas9 knockout led to recruitment and hyper-accumulation of RNF8 at the sites of DNA lesions induced by various sources of DNA damage, such as IR, UV or two-photon micro-irradiation. This RNF8 hyper-accumulation was observed at different time points after induction of DNA damage. We also observed an intact accumulation of factors and signals downstream of RNF8, including BRCA1 and K63-Ub, contrasting with findings reported by Pfeiffer *et al*. However, our results regarding defective recruitment of RNF168 and 53BP1 to sites of DNA damage after ATX3 inactivation were similar to those obtained by Pfeiffer *et al*.

In summary, our results suggest that ATX3 regulates p97-dependent RNF8 extraction from, but not recruitment to, sites of DNA lesions. Our findings are further supported by the fact that ATX3 inactivation selectively affects the NHEJ repair pathway, but not the HR repair pathway, and that IR-exposed HR-deficient, but not NHEJ-deficient, cancer cells are hypersensitive to ATX3 inactivation. Our results thus suggest that targeting the p97–ATX3 complex could be a promising strategy to radio-sensitise cancers, especially those that are defective in homologous recombination repair.

# Materials and Methods

## Contact for reagents and resource sharing

Further information and requests for reagents should be directed to the corresponding author: Kristijan Ramadan (kristijan.ramadan@oncology.ox.ac.uk).

## Experimental model and subject details

### Cell lines and cell culture procedure

The human HeLa (cervical cancer cell line), HEK293 and HEK293T (embryonic kidney cell lines), U2OS (osteosarcoma cell line), MCF7 (adenocarcinoma mammary epithelial cell line) and DLD1 (colorectal adenocarcinoma cell line) cells were cultured in Dulbecco's modified Eagle's medium (DMEM, high glucose, GlutaMAX) with 10% FBS and penicillin–streptomycin antibiotics under standard cell culture conditions. Cells were grown in a 5% $CO_2$ incubator at 37°C till 50–80% confluence before being used for experiments. Doxycycline-inducible Flp-In T-REx cell lines were maintained in hygromycin B (100 μg/ml)/blasticidin S (10 μg/ml)-containing medium, and the expression of proteins was induced with doxycycline (1 μg/ml) for 24 h. Cells were routinely tested for mycoplasma contamination. Sources of all the cell lines used in this study are provided as follows:

| | | |
|---|---|---|
| Human: osteosarcoma U-2 OS cells (female) | ATCC | Cat# HTB-96, RRID: CVCL_0042 |
| Human: embryonic kidney HEK293 cells (female) | ATCC | Cat# 300192/p777_HEK293, RRID: CVCL_0045 |
| Human: cervical carcinoma HeLa cells (female) | ATCC | Cat# CCL-2, RRID: CVCL_0030 |

| Human: CRISPR Δ-ATX3 HeLa cells | This study | N/A |
|---|---|---|
| Human: CRISPR Δ-ATX3 HEK293 cells | This study | N/A |
| Human: CRISPR Δ-ATX3 U2OS cells | This study | N/A |
| Human: HEK293-Flp-In T-REx ATX3-WT-nEGFP | This study | N/A |
| Human: HEK293-Flp-In T-REx ATX3-C14A-nEGFP | This study | N/A |
| Human: HEK293-Flp-In T-REx ATX3-UIM-nEGFP | This study | N/A |
| Human: HEK293-Flp-In T-REx ATX3-VBM-nEGFP | This study | N/A |
| Human: HEK293-Flp-In T-REx p97-EQ-cSSH | Meerang et al (2011) and Ritz et al (2011) | N/A |
| Human: U2OS-RNF8-WT-nEGFP | Mailand et al (2007) | N/A |
| Human: U2OS- RNF8-RING-nEGFP | Mailand et al (2007) | N/A |
| Human: CRISPR Δ-53BP1 MCF7 cells | Cuella-Martin et al (2016) | N/A |
| Human: CRISPR Δ-BRCA2 DLD1 cells | Zimmer et al (2016) | N/A |

### Generation of doxycycline-inducible Flp-In T-REx stable cell lines

Doxycycline-inducible p97EQ HEK293-Flp-In T-REx stable cell lines were generated as described perversely (Ritz et al, 2011). ATX3 variant (WT, C14A, UIM*) HeLa-Flp-In T-REx stable cell lines (Thermo Fisher Scientific) were generated according to the manufacturer's protocol for doxycycline-inducible expression of ATX3 variants. The ATX3-WT cDNA and mutants were amplified by PCR and cloned into pcDNA5/FRT/TO-nEGFP vector (containing N-terminal GFP-tag) using BamH1 and Xho1 (NEW ENGLAND BioLabs) restriction sites. The FRT/TO constructs were transfected with the pOG44 vector into Flp-In host HeLa cells for site-specific integration into the genome. The transformed cells were selected under hygromycin B (150 μg/ml) to generate ATX3 HeLa-Flp-In T-REx stable cell lines. The expression of ATX3 was confirmed by doxycycline induction (1 μg/ml) followed by Western blot analysis, and three different clones were selected for each variant.

### Generation of CRISPR/Cas9 ATX3-knockout cells

Commercially available CRISPR/Cas9 KO plasmid containing three guide RNAs targeting Ataxin 3, and a homology-directed repair plasmid containing a puromycin resistance cassette were purchased from Santa Cruz Biotechnology. Plasmids (2.5 μg each) were transfected into early-passage HEK293, HeLa and U2OS cells using FuGENE HD (Promega Corporation). After 72 h of transfection, a medium supplemented with puromycin was added to the cells. The puromycin concentrations (1.25 μg/ml for HEK293 cells, 0.6 μg/ml for HeLa cells and 1 μg/ml for U2OS cells) required to kill wild-type cells were determined by puromycin kill curve. After 72 h, the puromycin-containing medium was removed and cells were sorted using a cell sorter into single-cell populations on a 96-well plate. ATX3 expression was analysed by Western blotting. Three HEK293, three HeLa and three U2OS clones showing loss of all detectable ATX3 were selected for subsequent analysis.

### Bacterial strains

Bacterial strain Escherichia coli DH5a (Thermo Fisher Scientific; 18265-017) was used for plasmid amplification and E. coli Rosetta 2 (DE3) (Novagen; 71405-3) for expression of recombinant proteins.

### Antibodies

Antibodies used in this study are obtained as follows:

| p97 (Rabbit polyclonal) | Homemade | Freire & Ramadan Labs |
|---|---|---|
| ATX3 (Mouse monoclonal; C-1H9) | Merck Millipore | Cat# MAB5360; RRID: AB_2129339 |
| ATX3 (Rabbit ATX3 full-length polyclonal) | Homemade | Freire & Ramadan Lab |
| RNF8 (Rabbit polyclonal) | Proteintech | Cat# 14112-1-AP* |
| RNF8 (Rabbit polyclonal) | Homemade | Ramadan Lab |
| Phospho-γ-H2AX (Ser139) (Rabbit polyclonal) | Novus Biologicals | Cat# NB100-2280; RRID:AB_10000580 |
| Phospho- γ-H2AX (Ser139) | GeneTex | Cat# GTX127342* |
| 53BP1 (Rabbit polyclonal) | Santa Cruz Biotechnology | Cat# sc-22760; RRID: AB_2256326 |
| 53BP1 (Mouse monoclonal; C-19) | BD Biosciences | Cat# 612523; RRID:AB_399824 |
| HA (Rabbit polyclonal) | Santa Cruz Biotechnology | Cat# sc-805; RRID:AB_631618 |
| HA [clone 3F10] (Rat monoclonal) | Roche | Cat# 3F10; RRID:AB_2314622 |
| PCNA (Mouse monoclonal) | Abcam | Cat# ab29; RRID:AB_303394 |
| Vinculin (Mouse monoclonal; C-VIN54) | Abcam | Cat# ab130007; RRID: AB_11156698 |
| Flag (Rabbit polyclonal) | Sigma-Aldrich | Cat# F7425; RRID:AB_439687 |
| MDC1 (Mouse monoclonal; M2444) | Sigma-Aldrich | Cat# M2444, RRID:AB_532268 |
| Ufd1(Rabbit polyclonal) | Homemade | Freire & Ramadan Labs |
| Npl4 (Rabbit polyclonal) | Homemade | Freire & Ramadan Labs |
| RNF168 (Rabbit polyclonal) | Homemade | Freire & Ramadan Labs |
| K48-Ub (Rabbit monoclonal; C-Apu2) | Merck Millipore | Cat# 05-1307; RRID: AB_1587578 |
| K48-Ub (human monoclonal) | Genentech | C-Apu2.07 |
| K63-Ub (Rabbit monoclonal; C-Apu3) | Merck Millipore | Cat# 05-1308; RRID: AB_1587580 |
| K63-Ub (human monoclonal) | Genentech | C-Apu3.A8 |
| FK2-Ub (Mouse monoclonal; C-FK2) | Enzo Life Sciences | Cat# BML-PW8810; RRID:AB_10541840 |
| Proteasomal Subunit α6 | Enzo Life Sciences | Cat# BML-PW8195, RRID:AB_10541045 |

| | | |
|---|---|---|
| RAP80 (Rabbit polyclonal) | Bethyl | Cat# A300-763A; RRID: AB_669796 |
| α-Tubulin (Mouse monoclonal; DM1A) | Sigma-Aldrich | Cat# T6199; RRID:AB_477583 |
| β-Actin (Mouse monoclonal) | Abcam | Cat# ab6276; RRID:AB_2223210 |
| RAD51 (Rabbit polyclonal; H-92) | Santa Cruz Biotechnology | Cat# sc-8349; RRID: AB_2253533 |
| BRCA1 (Mouse monoclonal; C-D-9) | Santa Cruz Biotechnology | Cat# sc-6954; RRID:AB_626761 |
| GFP (Goat polyclonal) | Santa Cruz Biotechnology | Cat# sc-5385, RRID:AB_641121 |
| Cyclin E [clone HE12] (Mouse monoclonal) | Millipore | Cat# 05-363; RRID:AB_2071085 |
| Cyclin A [clone H-432] (Rabbit polyclonal) | Santa Cruz Biotechnology | Cat# sc-751; RRID:AB_631329 |
| Lamin B1 (Rabbit polyclonal) | Thermo Fisher Scientific | Cat# PA5-19468; RRID: AB_10985414 |
| Anti-BRCA2 (Ab-1) (Mouse monoclonal 2B) | Merck Millipore | Cat# OP95, RRID:AB_2067762 |
| Phospho-RPA32 (S4/S8) (Rabbit polyclonal) | Bethyl | Cat# A300-245A; RRID: AB_210547 |
| Rabbit IgG, HRP-conjugated (goat polyclonal) | Sigma-Aldrich | Cat# A9169; RRID:AB_258434 |
| Mouse IgG, HRP-conjugated (rabbit polyclonal) | Sigma-Aldrich | Cat# A9044; RRID:AB_258431 |
| Rat IgG, HRP-conjugated (goat polyclonal) | Bio-Rad | Cat# 5204-2504; RRID: AB_619913 |
| Alexa Fluor 488, Goat Anti-Mouse IgG (H+L) | Thermo Fisher Scientific | Cat# (A-11001); RRID: AB_2534069 |
| Alexa Fluor 488, Goat Anti-Rabbit IgG (H+L) | Thermo Fisher Scientific | Cat# A-11034; RRID: AB_2576217 |
| Alexa Fluor 594, Goat Anti-Mouse IgG (H+L) | Thermo Fisher Scientific | Cat# A-11020; RRID: AB_141974 |
| Alexa Fluor 594, Goat Anti-Rabbit IgG (H+L) | Thermo Fisher Scientific | Cat# A-11012; RRID: AB_2534079 |
| Alexa Fluor 488, Goat Anti-Human IgG (Fcγ fragment-specific) | Jackson ImmunoResearch Laboratories | Cat# 109-545-008; RRID:AB_2337833 |
| DyLight 488 goat anti-human | Thermo Fisher Scientific | Cat# SA5-10102; RRID: AB_2556682 |

*RRID for these antibodies could not be found.

## Plasmids

Plasmids were transfected using polyethyleneimine (PEI) reagent or FuGENE HD transfection reagent (Promega), and the medium was changed after 4 h of plasmid transfection to avoid the toxicity. The expression of plasmids was allowed for 24–48 h, and cells were harvested. In the case of U2OS cells, the plasmids were transfected with Lipofectamine 3000 reagent (Thermo Fisher Scientific) according to the manufacturer's protocol. All the plasmids used in this study are listed as follows:

| | | |
|---|---|---|
| pCDNA5-FRT/TO-nEGFP | Vaz et al (2016) | N/A |
| pFLAG-6a-Ataxin3Q22(Wt) | Addgene | Cat# 22126 |
| pFLAG-6a-Ataxin3Q22-C14A | Addgene | Cat# 22127 |
| pFLAG-6a-Ataxin3Q22-UIM* | Addgene | Cat# 22128 |
| pFLAG-6a-Ataxin3Q22-VBM* | This paper | N/A |
| pCDNA5-FRT/TO-ATX3-Wt-nEGFP | This paper | N/A |
| pCDNA5-FRT/TO-ATX3-C14A-nEGFP | This paper | N/A |
| pCDNA5-FRT/TO-ATX3-UIM-nEGFP | This paper | N/A |
| p3XFLAG-CMV-7.1 | Sigma-Aldrich | Cat# E7533 |
| pcDNA4/TO FLAG-RNF8 | Mailand et al (2007) | N/A |
| pcDNA4/TO FLAG-Ring* | Mailand et al (2007) | N/A |
| p3XFLAG-CMV-7.1_H1 | Gift from M. Gyrd-Hansen | N/A |
| pcDNA5 p97-Wt (Strep-myc)-amp | Meerang et al (2011) | N/A |
| pcDNA5 p97-EQ (Strep-myc)-amp | Meerang et al (2011) | N/A |
| pcDNA HA-Ubiquitin | Gift from H. Meyer | N/A |
| pCDNA5- FRT/TO-ATX3-VBM-nEGFP (RKRR-AAAA) | Ramadan Lab | N/A |

## siRNAs

The siRNA duplexes for the control (luciferase) or against the indicated gene were transfected using Lipofectamine RNAiMAX reagent (Thermo Fisher Scientific) according to the manufacturer's protocol, and depletion efficiency was assayed at 48–72 h post-transfection. All the siRNAs used in this study are listed as follows:

| | |
|---|---|
| siRNA_Luciferase (CGUACGCGGAAUACUUCGA) | Microsynth |
| siRNA_p97#1 (AACAGCCAUUCUCAAACAGAA) | Meerang et al (2011) |
| siRNA_p97#2 (AAGUAGGGUAUGAUGACAUUG) | Meerang et al (2011) |
| siRNA_ATX3#1 (GGACAGAGUUCACAUCCAU) | ON-Target; Dharmacon |
| siRNA_ATX3#2 (ACAGGAAGGUUAUUCUAUA) | ON-Target; Dharmacon |
| siRNA_ATX3#3 (GCAGGGCUAUUCAGCUAAG) | ON-Target; Dharmacon |
| siRNA_ATX3#4 (UGCGUCGGUUGUAGGACUAAA) | QIAGEN |
| siRNA_ATX3#5 (GGAAUGUAGGUGUCUGCUU) against 3'UTR | Microsynth |
| siRNF8 (GGACAAUUAUGGACAACAA) | Mailand et al (2007) |
| siRNF168 (GGCGAAGAGCGAUGGAAGA) | Doil et al (2009) |
| siDNA-PK (GCGUUGGAGUGCUACAACA) | ON-Target; Dharmacon |

| siRAD51 (GAAGCUAUGUUCGCCAUUA) (GCAGUGAUGUCCUGGAUAA) | ON-Target; Dharmacon |
| siNpl4 (CAGCCUCCUCCAAGAAAUC) | Meerang et al (2011) |
| siUfd1 (GUGGCCACCUACUCCAAAU) | Meerang et al (2011) |

## Preparation of cell lysates

Cells were washed twice with ice-cold PBS and subsequently lysed with RIPA buffer (25 mM Tris–HCl, pH 7.4, 150 mM NaCl, 0.1% SDS, 0.5% sodium deoxycholate, 1% Triton X-100) freshly supplemented with phosphatase and protease inhibitors (ROCHE). The deubiquitinating enzymes were also blocked with freshly prepared NEM (Thermo Scientific). Cell lysates were sonicated using the Bioruptor (Diagenode) at high intensity for 30 s on/30 s off cycles for 10 min to shear the DNA. Cell lysates were centrifuged at 14,000 $g$ for 10 min at 4°C, and the supernatant was collected.

## Cellular fractionation

With regard to specific subcellular fractions, the cells were fractionated as described previously (Mendez & Stillman, 2000). In brief, the cytoplasmic fraction was isolated by suspending cells in buffer A (10 mM HEPES, 10 mM KCl, 1.5 mM $MgCl_2$, 0.34 M sucrose, 10% glycerol) freshly supplemented with protease and phosphatase inhibitors, NEM and 0.1% Triton X-100. Nuclei were ruptured to release the nucleoplasm using hypotonic buffer B (10 mM HEPES, 3 mM EDTA, 0.2 mM EGTA, pH 8.0). The chromatin was washed with benzonase buffer (25 mM Tris–HCl, pH 8.0, 20 mM NaCl, 2 mM $MgCl_2$) and digested with benzonase (500 U/ml) while rotating at 10 rpm for 30 min at room temperature. Protein concentrations were measured using the RC DC Protein Assay Kit (Bio-Rad).

## Western blot

Cell lysates were denatured by boiling with Laemmli buffer containing beta-mercaptoethanol and separated on sodium dodecyl sulphate–polyacrylamide gel electrophoresis (SDS–PAGE) as per standard protocols. Proteins were transferred to PVDF (Bio-Rad) or nitrocellulose membrane (GE Healthcare) and detected by immunoblotting. The membranes were blocked with 5% milk and incubated overnight with primary antibodies (5% BSA in TBS–Tween buffer, pH 7.4) at 4°C. After three subsequent washings, secondary antibodies coupled with HRP were incubated for 1 h at room temperature and the specific protein band detected by ECL-based chemiluminescence with Bio-Rad gel documentation system.

## Ionising radiation

Cells were exposed to the indicated gray (Gy) of ionising radiation using a [137]caesium source (Gamma-Service Medical GmbH) at a dose rate of 1.3 Gy/min. Cells were left to recover in a 5% $CO_2$ incubator at 37°C for the recovery times as indicated.

## Two-photon absorbed laser spot-induced DNA damage

Cells were presensitised with Hoechst (10 μg/ml) for 30 min in Ibidi μ-Dish 35 mm high Glass Bottom dishes. Cells were treated with indicated siRNAs for 3 days then FluoroBrite DMEM was added and cells placed on heated stage 37°C, 5% $CO_2$. Single spot micro-irradiation was achieved using a 100-fs-pulse width laser, operating at an 80 MHz repetition rate (Mai-Tai DeepSee, Spectra Physics, UK) tuned to 750 nm, power set to 7.5–10 mW, the dwell time of laser was 1.58 μs. Power was measured before each irradiation experiment using a Coherent FieldMate with PM10 head. The 2-photon energy per single spot of 1.58 μs and 10 mW average power. Laser energy on the sample (spot) is typically < 16 nJ. Bleaching settings was used in Zeiss ZEN software and Zeiss LSM 710 was used at 63×/1.4 objective to image the cells. Images were analysed using FIJI, and RNF8-GFP recruitment was normalised to background levels in the irradiated cells. Recruitment was quantified relative to 0 s time point, which was set as 1 unit of relative fluorescent intensity.

## UV-laser micro-irradiation

Cells were line-micro-irradiated using a built in-house UV-laser system based around a UV-A (355 nm) Q-switched Nd:YAG laser (Minilite™ II) that produced pulses of 5 ns duration. The laser beam was expanded by a 10× beam expander (Edmund #68-272) and passed through a bandpass filter (Semrock FF01-360/12-25) and an nominal OD1 reflective filter (Thorlabs NDUV10B) and then focused onto a slit with a cylindrical lens (Elliott CYLX-25.4X25.4U300, $f$ = 300 mm). The slit passed the required laser energy and was re-imaged onto the sample with a tube lens (Thorlabs LA4855-UV) and a 10×/0.5 objective (Nikon CFI Super Fluor). The slit height was adjusted such that the slit covered the imaged field of view and its width adjusted to provide a line width of ~ 450 nm fullwidth at half-maximum. The microscope body (Nikon TE2000) was modified to accept the additional tube lens, and imaging was performed using the TE2000's $f$ = 200 mm tube lens onto a sCMOS camera (Hamamatsu Orca Flash4.0). Cells were seeded onto 10-mm No. 1 cover glasses (VWR) and sensitised using 10 μg/ml Hoechst 33258 (Sigma) added 30 min prior to laser illumination. Coverslips were subsequently transferred into in-house-made glass-bottom dishes containing complete medium. Each laser line was generated with a single pulse, and the sample stage was moved by 25 μm after every pulse using a motorised stage (Märzhäuser SCAN IM). Typically, the imaged field could be covered by 50 laser lines. Laser energy pulse–pulse reproducibility was monitored by an in-house energy sensor based around pyroelectric (Scitec PEO8) sensor and a CFQ-1520 quartz coverslip pick off. Laser energy on the sample was adjusted on the laser's variable attenuator to be typically ≪ 10 μJ across the line. Following laser irradiation, cover glasses were transferred in 12-well plates containing complete medium and incubated in a 5% $CO_2$ incubator at 37°C for indicated recovery times.

## Colony formation assay

Cell survival after ionising radiation was determined by slandered clonogenic assay. In brief, appropriate numbers of cells were seeded in 6-well plates and incubated overnight. Cells were exposed to the

indicated dose of ionising radiation and incubated in a 5% $CO_2$ incubator at 37°C for 7–10 days to form colonies (at least 50 cells per colony). The colonies were fixed and stained with crystal violet (Sigma) solution in methanol for 20 min at room temperature. After washing and drying, the colonies were counted manually or by automated colony counter GelCount™ (DTI-Biotech). Results were normalised to plating efficiency, and the number of colonies in irradiated samples was expressed as a percentage of non-irradiated samples.

### Immunofluorescence

Immunofluorescence was performed as previously described (Meerang *et al*, 2011). Briefly, cells were fixed with 4% PFA in PBS for 15 min at RT followed by permeabilisation with 0.5% Trion X-100 (Sigma). If necessary, pre-extraction was performed for 2 min on ice using pre-extraction buffer (25 mM HEPES pH 7.4, 50 mM NaCl, 1 mM EDTA, 3 mM $MgCl_2$, 300 mM sucrose, 0.5% Triton X-100, protease and phosphatase inhibitors) prior to fixation. Cells were blocked with 5% BSA overnight at 4°C. Antibodies were diluted in 2.5% BSA in PBS at specific dilutions and left for 1 h at room temperature. DNA was stained with DAPI and cells were mounted with FluoroMount-G hard mounting medium (Interchim).

### Microscopy and image quantifications

Nikon 90i epifluorescent microscope equipped with a Nikon DS-Qi1Mc digital microscope camera was used to take the entire IF images. Images were captured using a 60× objective and exported as a TIFF file. The Image analysis was performed using ImageJ software. For laser line analysis, intensity at the site of DNA damage was analysed by circling the γ-H2AX signal and measuring the intensity of the protein of interest at that area. From the same nucleus, a measurement of the background signal was also taken and subtracted from the intensity of the damage area. All measurements were normalised to control for each time points to compare between different siRNA-depleted conditions.

### Protein co-immunoprecipitations (Co-IPs)

Cells were transfected with the plasmids of interest using PEI. After treatment (according to the experiment), cells were collected by trypsinisation and rinsed twice with ice-cold PBS. Cells were lysed using IP lysis buffer (50 mM Tris–HCl pH 7.4, 150 mM NaCl, 1 mM EDTA, 0.5–1% Triton X-100, protease and phosphatase inhibitors). The pellets were washed once with benzonase buffer (2 mM $MgCl_2$, 50 mM Tris–HCl pH 8.0, 20 mM NaCl, Inhibitors) and then incubated in benzonase buffer containing benzonase (500 U/ml) at 4°C until the DNA was digested. Lysates were loaded onto beads (GFP: Chromotek; Flag: Sigma; Strep-Tactin: IBA) and capturing was performed according to the manufacturer's protocol. Proteins were eluted in Laemmli sample buffer (2×) and analysed by Western blotting.

### Denaturing immunoprecipitations (Denaturing-IPs)

Cells were transfected with the plasmids of interest using PEI or FuGENE HD Transfection Reagent (Promega). After treatment (according to the experiment), cells were collected by trypsinisation

and rinsed twice with ice-cold PBS. Cells were lysed with 1% SDS and heated to 95°C for 5 min to denature the proteins. The excess of SDS was quenched with 1% Triton X-100 and the DNA was sheared with a 22-G needle. Cell lysate was centrifuged at high speed to remove the cell debris and the supernatant was incubated with Flag beads at 4°C for 2–4 h. The beads were washed one time with 1 M NaCl/1% Triton X-100 followed by two washes with 150 mM NaCl/1% Triton X-100 and the bound protein was eluted with Laemmli sample buffer (2×) and analysed by Western blotting.

### SILAC and mass spectrometry

For quantitative mass spectrometry, Flp-In T-REx HEK293T/p97-E578Q-Strep cell lines were first SILAC labelled. After induction of p97-E578Q-Strep with doxycycline and treatment with 10 Gy of ionising radiation, a nuclear fraction was extracted, treated with benzonase, and the p97 interactors in this fraction were pulled down with Strep-Tactin beads (IBA). After purification, pulled-down proteins were prepared for mass spectrometry including digestion with Trypsin (Promega), and analysed by LC-MS/MS, followed by determination of SILAC ratios by Proteome Discoverer software (Thermo). Results from a subgroup of proteasomal and p97-cofactor proteins were selected for this study.

### Cloning and mutagenesis

ATX3-WT, C14A and UIM* variants cDNAs were amplified by Accu-Prime Pfx DNA polymerase (Invitrogen) using Veriti 96-well thermal cycler (Applied Biosystems). Plasmids pFLAG-6A-ATX3Q_22 (WT), C14A and UIM* from Addgene were used as templates to amplify the cDNAs. The cDNAs were cloned in pCDNA5-FRT/TO-nEGFP vector using BamH1 and Xho1 restriction endonucleases (NEW ENGLAND BioLabs) and confirmed by DNA sequencing (Source Biosciences). ATX3 variants were generated by site-directed mutagenesis using AccuPrime Pfx DNA polymerase (Invitrogen) according to the manufacturer's protocol and confirmed by DNA sequencing (Source Biosciences). List of PCR and mutagenic primers is provided as follows.

| | |
|---|---|
| ATXN3_BamH1_F (ATATGGATCCATGGAGTC CATCTTCCACGAG) | This study (PCR) |
| ATXN3_Xho1_R (ATATCTCGAGTTATT TTTTCCTTCTGTTTTCAAATCATTTCTG) | This study (PCR) |
| ATXN3_VBMRKRR285AKAR_R (ATCAGGT ACAAATCTTACTTCAGAAGAGCT TGCGAAGGCACGAGAAGCCTACTTT) | This study (mutagenesis) |
| ATXN3_VBMAKAR285AAAA_F (CTTCAGAAGAGCTTGCGGCGGCAGCAG AAGCCTACTTTGAAAAAC) | This study (mutagenesis) |
| ATXN3_VBMAKAR285AAAA_R (GTTTTTCAAAGTAGGCTTCTGCTGCCG CCGCAAGCTCTTCTGAAG) | This study (mutagenesis) |

### Reverse transcription–polymerase chain reaction

Total RNAs were isolated from cells using GeneJET RNA purification kit (Thermo Fisher Scientific; K0731) and converted to cDNA

using SuperScript™ II Reverse Transcriptase (Invitrogen; 18064014). Reverse transcription–polymerase chain reaction (RT–PCR) was performed on StepOnePlus RT–PCR system (Applied Biosystem) with gene-specific primers using Fast SYBR Green Master Mix (Thermo Fisher Scientific; 4385612). List of gene-specific primers used for RT–PCR in this study is provided as follows:

| | | |
|---|---|---|
| Human_ATXN3_ Forward RT Primer (GCACGTTTTTACAGCAGCCTT) | This study (RT–PCR) | |
| Human_ATXN3_ Reverse RT Primer (AGCCTCTGATACTCTGGACTGT) | This study (RT–PCR) | |
| Human_RNF8_ Forward RT Primer_P1 (TTGAGGCTGTCACCTTGAACT) | This study (RT–PCR) | |
| Human_ RNF8_ Reverse RT Primer_P1 (TGTCCTTCCGACAAATGGGG) | This study (RT–PCR) | |
| Human_ RNF8_ Forward RT Primer_P2 (TCGCAGCAAGAAGGACTTTGA) | This study (RT–PCR) | |
| Human_ RNF8_ Forward RT Primer_P2 (AGTTCAAGGTGACAGCCTCAAT) | This study (RT–PCR) | |

## Comet assay

HeLa cells were treated with 20 nM non-targeting control siRNA (Qiagen) or ATXN3 siRNA for 48 h, trypsinised and irradiated in suspension at 4 Gy using a CellRad X-ray irradiator (Faxitron). Cells were embedded in agarose on a microscope slide, incubated for up to 4 h in a humidified chamber and levels of DNA double-strand breaks measured using the neutral comet assay, as previously described (Nickson *et al*, 2017; Carter *et al*, 2018).

## Reporter assay

DSB repair efficiency was measured as described previously (Bennardo *et al*, 2008). In brief, $0.2 \times 10^6$ cells were seeded and treated with different siRNAs. The next day cells were transfected with the pCBASceI plasmid and allowed to grow for 4 days. GFP was measured by flow cytometry performed on a FACSCalibur instrument (BD Biosciences).

## Other reagents

List of chemicals, peptides and recombinant proteins used in this study is provided as follows:

| | | |
|---|---|---|
| CB-5083 | Selleckchem.com | Cat# S8101 |
| NMS873 | Selleckchem.com | Cat# S7285 |
| DBeQ | Selleckchem.com | Cat# S7199 |
| Cycloheximide | Sigma | Cat# C1988 |
| MG132 | Selleckchem.com | Cat# S2619 |
| 5-Ethynyl-2′-deoxyuridine (EdU) | Sigma-Aldrich | Cat# T511285 |
| 5-Bromo-2′-deoxyuridine (BrdU) | Sigma-Aldrich | Cat# B9285 |
| Ampicillin | Fisher Scientific | Cat# BP1760 |

| | | |
|---|---|---|
| Blasticidin S | Thermo Fisher Scientific | Cat# A1113903 |
| Hygromycin B | TOKU-E | Cat# SKU: H007 |
| Doxycycline | PanReac AppliChem | Cat# A2951,0025 |
| AccuPrime *Pfx* DNA polymerase | Invitrogen | Cat# 12344-024 |
| Lipofectamine siRNAMax | Invitrogen | Cat# 13778-150 |
| Polyethylenimine | Sigma-Aldrich | Cat# 408727 |
| Benzonase | Millipore | Cat# 71205 |
| Kanamycin | Thermo Fisher Scientific | Cat# 11815024 |
| Strep-Tactin Sepharose | IBA | Cat# 2-1201-010 |
| Penicillin–Streptomycin | Sigma | Cat# P4333 |
| ANTI-FLAG® M2 Affinity Gel | Sigma | Cat# A2220 |
| GFP-Trap®_A | Chromotek | Cat# GTA-500 µl |
| NEM | Sigma-Aldrich, Thermo Fisher Scientific | Cat# E3876-25GNEM |
| FuGENE HD Transfection Reagent | Promega | Cat# E2311 |

## Critical commercial assays

List of critical chemical assays used in this study is provided as follows:

| | | |
|---|---|---|
| Clarity™ Western ECL Substrate | Bio-Rad | Cat# 1705061 |
| SuperSignal™ West Femto Maximum Sensitivity Substrate | Thermo Fisher Scientific | Cat# 34095 |
| SuperSignal™ West Pico PLUS Chemiluminescent Substrate | Thermo Fisher Scientific | Cat# 34580 |
| *DC*™ Protein Assay Kit II | Bio-Rad | Cat# 5000112 |
| Plasmid DNA Purification Kit | QIAGEN | Cat# 12163 |
| Gel Extraction Kit | QIAGEN | Cat# 28604 |
| PCR Purification Kit | QIAGEN | Cat# 28104 |
| QuikChange II Site-Directed Mutagenesis Kit | Agilent | Cat# 200523 |
| GeneJET RNA purification kit | Thermo Fisher Scientific | Cat# K0731 |
| Fast SYBR Green Master Mix | Thermo Fisher Scientific | Cat# 4385612 |
| SuperScript™ II Reverse Transcriptase | Thermo Fisher Scientific | Cat# 18064014 |

## Quantification and statistical analysis

Statistical analysis was performed using GraphPad Prism software. Unpaired *t*-tests and one- or two-way ANOVAs were performed as

mentioned in the Figure legends. *P* values refer to: ****$P < 0.0001$, ***$P < 0.001$, **$P < 0.01$, *$P < 0.05$, $^{ns}P > 0.05$. For IF micrographs the scale bar is 10 μm and at least 100 nuclei were quantified per condition per experiment unless mentioned.

**Expanded View** for this article is available online.

## Acknowledgements

The Ramadan Laboratory is supported by a Medical Research Council UK programme grant (MC_EX_MR/K022830/1) to K.R. J.O. was supported by a Swiss National Science Foundation grant (31003A_141197) to K.R., a Goodger and Schorstein Scholarship, University of Oxford, and a European Institute of Innovation and Technology short-term post-doctoral fellowship. S. K, J. F. and S. L were supported by the Cancer Research UK or MRC studentships. B. Vo., A. E. K. and I. D. C. T. are supported by Cancer Research UK Programme Grant C5255/A15935. C.G. was supported by the Danish Cancer Society (Grant No. R146-RP11394). R. F. is supported by Spanish Ministry of Innovation Science and Universities/EU-ERDF (SAF2016-80626-R), E. C. is supported by the Fundación DISA, and E. H-C. is supported by an ACIISI pre-doctoral fellowship. We thank Ross Chapman for providing us with WT and 53BP1-deficient MCF7 cell lines. We are grateful to Pam Reynolds for support with the UV micro-irradiation system, to the Mechanical and Electronics Workshops (J Prentice and RG Newman) for assistance with construction of the system, to the Bioanalysis Core (Graham Brown and Ioland Vendrell) for the supports with microscopy and mass spectrometry and to the Radiation Biophysics Core for the help with X-ray radiation. We thank Benedikt Kessler for his constant support with mass spectrometry. We thank Jeremy Stark for providing us with GFP-reporter assays for HR and NHEJ.

## Author contributions

JO and ANS performed the majority of the experiments. IT performed a SILAC-based p97 proteome. SK and AEK performed and analysed end-resection assay. SL performed two-photon laser-induced DNA damage live imaging. BVa analysed DNA repair reporter assays. CG and NM investigated GFP-RNF8 foci formation, and JF created ATX3-ko cells. IDCT, JO, PRB and BVo designed, optimised and calibrated in-house UV-laser micro-irradiation system. EH-C, EC and RF performed cell cycle analysis and created rabbit anti-human ATX3 antibody. JP performed comet assay, and MM initially observed the role of p97 in the stability of RNF8 and contributed with Fig 1C. KR conceived and supervised the project. JO, ANS and KR jointly wrote and prepared the manuscript. All the authors discussed the results.

## Conflict of interest

The authors declare that they have no conflict of interest.

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
