## [Review Process File · The EMBO Journal]

The p97-Ataxin 3 complex regulates homeostasis of the DNA damage response E3 ubiquitin ligase RNF8

Abhay Narayan Singh, Judith Oehler, Ignacio Torrecilla, Susan Kilgas, Shudong Li, Bruno Vaz, Claire Guérillon, John Fielden, Esperanza Hernandez-Carralero, Elisa Cabrera, Iain D.C. Tullis, Mayura Meerang, Paul R Barber, Raimundo Freire, Jason Parsons, Borivoj Vojnovic, Anne E. Kiltie, Niels Mailand and Kristijan Ramadan.

Review timeline:

Submission date:	5 th May 2019
Editorial Correspondence:	6 th June 2019
Author correspondence:	19 th June 2019
Editorial Decision:	25 th June 2019
Revision received:	16 th September 2019
Accepted:	19 th September 2019

Editor: Hartmut Vodermaier

Transaction Report:

Editorial Correspondence

6th June 2019

Thank you for your patience during our arbitration on your manuscript on RNF8-p97-ATX3, submitted to us with reports from previous review at another journal. I sent it to three trusted referees of our own journal, one of whom (ref 1) was also involved in the evaluation of the earlier work by Pfeiffer et al, and one (ref 3) with particular expertise in aspects of p97 function. I asked them to particularly look at how you addressed the previously raised concerns, with special focus on their respective expertise/familiarity. As you will see from the comments below, arbitrators 1 and 2 are for the most part happy with the study in its present form, but referee 3 does feel that some of the original criticisms would still seem valid, and also raises some specific issues themselves. In light of the positive opinions of the other two referees, I would in this case not argue for omission of the first part of the study, and also not consider a full-blown follow-up investigation on the second part absolutely essential; however, I do think it would be important to address some of the most immediate and salient issues, in particular with regard to p97 biology, before publication in The EMBO Journal would be warranted. To better determine what could at this stage be done to address the issues raised by arbitrating referee 3, I would like to give you the option to consider the enclosed reports, and to respond back to me with a tentative point-by-point response letter in the coming days. Based on this, we can then further discuss proceedings (possibly also directly by phone).

REFeree REPORTS

Referee #1:

You asked me to look over this manuscript and its review process with regard to the previous

Dantuma paper (also on ATX3 and the DDR).

I really think the Ramadan group have done all they reasonably could to tackle the discrepancy with the previous report - including asking another lab to look at it.

Why the difference exists I'm not sure - but its worth remembering that the impact Dantuma reported on MDC1 was slight and needed sophisticated analysis.

As regard answering the reviewers again I think they are there. While some minor elements might be nice to see (RNF8 foci time course for example on ATX depletion in G1 Vs S/G2 cells) I doubt the story over-all would be much improved.

Referee #2:

I was asked to comment on the manuscript and prior response to reviews. Overall, this is an interesting study and helps clarify differences between RNF8 and RNF168. It also has some murky areas, which is perhaps not unexpected given the myriad processes that p97 is involved with. I am generally convinced by the experimental evidence, however think that it would be valuable to use more knockout lines rather than double siRNA to perform epistasis experiments. These are less convincing and could be performed using better tools.

Referee #3:

The manuscript by Singh et al. contains an enormous amount of data, and this reviewer appreciates its difficult history in light of a competing publication by the Dantuma lab (Pfeiffer et al, 2017). However, this reviewer also shares the criticism raised by previous reviewers that the manuscript combines two loosely connected stories about the control of RNF8 by p97 and ATX3 that are difficult to reconcile.

The first part reports that ATX3 antagonizes the p97-dependent proteasomal degradation of RNF8. This could simply reflect a house-keeping quality control function of p97 and ATX3 and, as such, would not be particularly original given a number of previous, similar observations for other p97-ATX3 substrates. In order to make this part suitable for publication in EMBOJ, a significant number of additional experiments leading to substantial mechanistic insights into the role of ATX3 in p97-mediated substrate turnover would be required (probably also including a reconstituted *in vitro* assay), in addition to consolidating the existing data (see specific points below).

Alternatively, the first part could be largely or completely omitted, and the manuscript could be re-focused on the much more interesting second part, i.e. the role of p97 and ATX3 in controlling RNF8 level and activity at the sites of DSBs. This second part contains a comprehensive and for the most part highly convincing description of the involvement of ATX3 in DSB repair by NHEJ, but not HR, and thus in repair pathway choice. This story should still be strengthened in two ways: First, the exact roles of ATX3 and RNF8 in the process should be discussed in more detail. E.g., is RNF8 at DSBs auto-ubiquitylated with K63 or K48 chains? (The accumulation of RNF8 and K63-Ub, but not K48-Ub upon depletion of ATX3 would suggest the first possibility.) Why does the accumulation of RNF8 upon depletion of ATX3 result in a counterintuitively reduced ubiquitination of its substrate, histone H1? Why/how would other substrates "suppress the timely ubiquitination of H1" (p. 15) when its E3 ligase RNF8 is present in excess? What is the role of UFD1-NPL4 in RNF8 turnover at DSBs? (A potential accumulation of RNF8 upon depletion of UFD1 or NPL4 was not tested.) Second, the key hypothesis that H1 ubiquitination is impaired because of loosened chromatin and/or dissociation of H1 from the vicinity of DSBs should be confirmed experimentally, e.g. by H1 and CHD4 staining at DSBs by IF or by appropriate PLA experiments. (The Figure B for reviewers shows total chromatin of unstressed cells, which may be irrelevant for the vicinity of DSBs. Also, the data are not really convincing, as unequal total amounts of chromatin appear to be loaded in lane 2 vs 5.)

Regarding the results contradicting the previous paper by Pfeiffer et al, the authors present convincing microscopy data supporting their conclusions. However, as to the question if MDC1 is a target/interactor of ATX3, the negative result in Fig. S4G is not really convincing, given that the IP of GFP-ATX3 is very inefficient (depletion rather than enrichment in comparison to the input

levels). To strengthen their point, the authors should either provide a better IP of GFP-ATX3, or probe the Flag-IPs in Fig. 3D with a MDC1 antibody. Moreover, it might be advantageous to discuss all consensus and discrepancies with Pfeiffer et al in one paragraph of the Discussion section, rather than spreading it over several places.

Specific points:

1. While the CHX chase assays monitor the degradation of endogenous RNF8, all experiments in Figs. 1 through 4 addressing the ubiquitination state of RNF8 and its interaction with p97 and ATX3 were performed with cells strongly overexpressing RNF8, p97 and/or ATX3. Since overexpression can lead to completely erroneous conclusions, at least the key results must be confirmed at endogenous expression levels. For instance, the HA blots of the inputs in Figs. 2A and 2C show that overexpression of FLAG-RNF8 leads to a massive increase of total (!) ubiquitinated proteins in the lysate, which is a completely artificial setting.

2. Please note that the co-IP results do not support the existence of a "p97-ATX3-RNF8-(proteasome) complex" (pp. 8, 18), since the co-IPs could equally well reflect binary interactions.

3. Figs. 1F, 2AC, 4BC, 6C: The conditions used for the denaturing IPs (1% SDS) will certainly denature most if not all proteins in the lysates, but will not prevent subsequent unspecific interactions of "sticky" proteins, also including (partially) denatured ubiquitin chains. This leaves the possibility that not all ubiquitinated species detected in these experiments are RNF8 conjugates. Key findings need to be confirmed using His-Ub expressing cells in fully denaturing (GdnHCl/urea) Ni-NTA pull-down experiments, preferably detecting endogenous RNF8 and H1, respectively. Regarding Fig. 6C: Was Flag-H1 isolated from total cell lysates or from a chromatin-enriched fraction? The latter would be the appropriate experiment.

4. The proteomics data summarized in Fig. 3A and the corresponding experimental methods have not been subject to peer review. Consequently, the authors should either show all necessary details and results, or remove the figure and just refer to the data as unpublished results in the text.

5. NMS-873 does not disrupt the hexameric state of p97 (p. 9).

We would like to thank the referees on their comments and suggestions on our manuscript entitled “**Homeostasis of the E3-ubiquitin ligase RNF8 by the p97-Ataxin 3 complex preserves genome stability by regulating DNA repair pathway choice**” (EMBO J-2019-102361).

We are very happy to hear that referee 1 and 2 are completely satisfied with our revised version of the manuscript prepared for another journal. However, referee 3, although convinced and satisfied with our manuscript, raises some specific issues. We believe we can easily address referee 3's concerns either by clarifying certain aspects or experimentally.

Specifically,

Referee#1

You asked me to look over this manuscript and its review process with regard to the previous Dantuma paper (also on ATX3 and the DDR). I really think the Ramadan group have done all they reasonably could to tackle the discrepancy with the previous report - including asking another lab to look at it. Why the difference exists I'm not sure - but its worth remembering that the impact Dantuma reported on MDC1 was slight and needed sophisticated analysis. As regard answering the reviewers again I think they are there. While some minor elements might be nice to see (RNF8 foci time course for example on ATX depletion in G1 Vs S/G2 cells) I doubt the story over-all would be much improved.

Answer 1: Thank you. We highly appreciate your opinion and support of this manuscript.

Referee#2

I was asked to comment on the manuscript and prior response to reviews. Overall, this is an interesting study and helps clarify differences between RNF8 and RNF168. It also has some murky areas, which is perhaps not unexpected given the myriad processes that p97 is involved with. I am generally convinced by the experimental evidence, however think that it would be valuable to use more knockout lines rather than double siRNA to perform epistasis experiments. These are less convincing and could be performed using better tools.

Answer 2: Thank you. We highly appreciate your opinion and suggestion to use more knockout lines. We also believe that ATX3 and RNF8 epistasis would be much better to analyse in double ATX3/RNF8 knock out cells. However, RNF8 is a fitness gene in all tested human cell lines. Thus, human cells would not survive RNF8 knock out or it would adapt on RNF8 lost in ATX3-knock out cells and thus mask here observed epistatic phenotype.

Referee#3

The manuscript by Singh et al. contains an enormous amount of data, and this reviewer appreciates its difficult history in light of a competing publication by the Dantuma lab (Pfeiffer et al, 2017). However, this reviewer also shares the criticism raised by previous reviewers that the manuscript combines two loosely connected stories about the control of RNF8 by p97 and ATX3 that are difficult to reconcile. The first part reports that ATX3 antagonizes the p97-dependent proteasomal degradation of RNF8. This could simply reflect a house-keeping quality control function of p97 and ATX3 and, as such, would not be particularly original given a number of previous, similar observations for other p97-ATX3 substrates. In order to make this part suitable for publication in EMBOJ, a significant number of additional experiments leading to substantial mechanistic insights into the role of ATX3 in p97-mediated substrate turnover would be required (probably also including a reconstituted in vitro assay), in addition to consolidating the existing data (see specific points below). Alternatively, the first part could be largely or completely omitted, and the manuscript could be re-focused on the much more interesting second part, i.e. the role of p97 and ATX3 in controlling RNF8 level and activity at the sites of DSBs.

Answer 3: Thank you for your opinion and suggestions. As discussed at several places throughout the manuscript, the regulation of RNF8 homeostasis at protein level is essential for tumorigenesis, cancer cell proliferation and facilitates cancer chemoresistance and poor patient prognosis (Lee HJ

et al., Mol. Cell, 2016, Wang S et al., Biochim Biophys Acta Mol Basis Dis. 2017, Li L et al., J Clin Invest. 2018). These reports suggest that either high or low RNF8 protein expression leads to cancer development and poor patient prognosis.

Also, a recent report nicely demonstrates the role of RNF8 in DNA replication (Schmid et al., Mol Cell, 2018). However, how RNF8 at protein level, one of the fitness human genes, is regulated is not known. Thus, we believe our discovery of the p97-ATX3 complex as one of the main regulators of RNF8 homeostasis under unchallenged/physiological condition is essential for both DNA replication and cancer.

We agree that this part (RNF8 regulation under unchallenged conditions) of the manuscript could be a separate manuscript and it was highly appreciated by original reviewer 3 from another journal. However, we believe it is important to report that the p97-ATX3 complex is the main regulator of RNF8 homeostasis under both unchallenged and genotoxic condition at one place. This is the main message of this manuscript.

...This second part contains a comprehensive and for the most part highly convincing description of the involvement of ATX3 in DSB repair by NHEJ, but not HR, and thus in repair pathway choice. This story should still be strengthened in two ways: First, the exact roles of ATX3 and RNF8 in the process should be discussed in more detail. E.g., is RNF8 at DSBs auto-ubiquitylated with K63 or K48 chains? (The accumulation of RNF8 and K63-Ub, but not K48-Ub upon depletion of ATX3 would suggest the first possibility.)

Answer 4: We show that RNF8 catalytically inactive variant (RING finger mutation in C403S) stuck and hyper-accumulates at the sites of DNA damage (Fig. S4 K and L) suggesting RNF8 without its own E3-ligase activity cannot be removed. Furthermore, depletion of ATX3 results in the increase K63-ubiquitin signal on chromatin that is completely abolished by RNF8 co-depletion (Figure 5 I and J). In contrary, the p97-ATX3 complex regulates preferentially K48-ubiquitin signal on total RNF8 (Figure 2A and 4C, isolated under denaturing conditions). However, RNF8 is preferentially ubiquitinated by K48-ubiquitin signal in both soluble and chromatin fraction (Fig. S1D). RNF8- K63-ubiquitin is also visible on both soluble and chromatin fraction but to a much lesser extent when compared to K48-ubiquitination. However, the affinity of specific K48- or K63-ubiquitin chain antibodies are similar (Figure S1E).

Thus, as discussed in our manuscript, we do not understand yet why ATX3 has different preferences in the regulation of soluble RNF8 vs chromatin bound RNF8. Different post-translational modifications on ATX3 after IR (such as phosphorylation) might regulate ATX3 preferences for either soluble or chromatin RNF8. For example, ATX3 has a SUMO Interacting domain (SIM) and it was shown that ATX3 is recruited to sites of DNA lesion by SUMO signal (Pfeiffer et al. EMBO J, 2017). So, it is possible that chromatin bound RNF8 is also SUMOylated and both ubiquitin and SUMO interaction between ATX3 and RNF8 regulates different outcome of RNF8 on chromatin vs soluble RNF8. These important questions could not be addressed in this manuscript as a separate in-depth analysis is required to address it properly.

Why does the accumulation of RNF8 upon depletion of ATX3 result in a counterintuitively reduced ubiquitination of its substrate, histone H1? Why/how would other substrates "suppress the timely ubiquitination of H1" (p. 15) when its E3 ligase RNF8 is present in excess?

Answer 5: We discussed this question on pages 20 and 21 (discussion part). Also, the recent finding suggests that histone H1 is even not the main substrate of RNF8, if at all, but L3MBTL2 (Nowsheen S, Nat Cell Biology 2018). It seems there is still an open question in the field, namely what is the main substrate of RNF8 at sites of DNA damage that is important for the propagation of the ubiquitin signal. Our manuscript due to its different focus - *regulation of RNF8 by the p97-ATX3 system* - could not address this important question in the field: What is the main RNF8 substrate at the sites of DNA damage?

What is the role of UFD1-NPL4 in RNF8 turnover at DSBs? (A potential accumulation of RNF8 upon depletion of UFD1 or NPL4 was not tested.

Answer 6: We will test the effect of Npl4-Ufd1 inactivation in RNF8 turnover at DSBs.

Second, the key hypothesis that H1 ubiquitination is impaired because of loosened chromatin and/or dissociation of H1 from the vicinity of DSBs should be confirmed experimentally, e.g. by H1 and CHD4 staining at DSBs by IF or by appropriate PLA experiments. (The Figure B for reviewers shows total chromatin of unstressed cells, which may be irrelevant for the vicinity of DSBs. Also, the data are not really convincing, as unequal total amounts of chromatin appear to be loaded in lane 2 vs 5.)

Answer 7: impaired H1-ubiquitination in ATX3-inactivated cells is not the key hypothesis of this manuscript, but that the p97-ATX3 complex regulates the main and first E3-ubiquitin ligase RNF8 in DNA damage response. How RNF8 regulates other substrates is not well understood in the literature and further work will be required (see answer 5). Unfortunately, this manuscript could not address this question. Thus, in this already complex manuscript that contains an enormous amount of data (stated by this referee), we cannot additionally analyse how the ATX3-RNF8 complex regulates chromatin status at sites of DNA damage.

Regarding the results contradicting the previous paper by Pfeiffer et al, the authors present convincing microscopy data supporting their conclusions. However, as to the question if MDC1 is a target/interactor of ATX3, the negative result in Fig. S4G is not really convincing, given that the IP of GFP-ATX3 is very inefficient (depletion rather than enrichment in comparison to the input levels). To strengthen their point, the authors should either provide a better IP of GFP-ATX3, or probe the Flag-IPs in Fig. 3D with a MDC1 antibody. Moreover, it might be advantageous to discuss all consensus and discrepancies with Pfeiffer et al in one paragraph of the Discussion section, rather than spreading it over several places.

Answer 8: We did multiple times this experiment and we will provide an additional GFP-ATX3 co-immunoprecipitation blot where ATX3 has efficiently been isolated. In all these experiments MDC1 was not present in the complex with ATX3.

Regarding discussing discrepancies with Pfeiffer et al, we can consider moving this part to the discussion section.

Specific points:

1. While the CHX chase assays monitor the degradation of endogenous RNF8, all experiments in Figs. 1 through 4 addressing the ubiquitination state of RNF8 and its interaction with p97 and ATX3 were performed with cells strongly overexpressing RNF8, p97 and/or ATX3. Since overexpression can lead to completely erroneous conclusions, at least the key results must be confirmed at endogenous expression levels. For instance, the HA blots of the inputs in Figs. 2A and 2C show that overexpression of FLAGRNF8 leads to a massive increase of total (!) ubiquitinated proteins in the lysate, which is a completely artificial setting.

Answer 9: The rescue of endogenous RNF8 degradation in ATX3-depleted cells with either ATX3-wt expression but not with ATX3-VBM variant (p97-binding deficient) (Fig. 4D) or by proteasome inhibitor MG132 (Fig. 2B), suggests that endogenous RNF8 is regulated by the ubiquitin signal. On top of it, we clearly demonstrated that over-expressed RNF8wt but not its E3-ligase defective variant (RNF8-RING) is prone to degradation (Figure 2D and compare with Fig. 2B). Altogether suggests that endogenous RNF8 is degraded in the ubiquitin dependent manner (proteasome and p97-dependent degradation) and that over-expressed RNF8 follows the similar degradation rule and kinetics as endogenous RNF8.

I agree, this could be also confirmed by isolating endogenous RNF8 and then check the ubiquitination status of RNF8. However, the majority of published manuscripts when analyse the ubiquitination status of specific protein use over-expressed proteins as read out as we did here for RNF8.

In summary, as we demonstrated that endogenous RNF8 degradation is similar to exogenous RNF8 degradation and this process depends on the proteasome, p97 and the E3-ligase activity of RNF8, we think it is not necessary to additionally demonstrate that endogenous RNF8 is also ubiquitinated.

2. Please note that the co-IP results do not support the existence of a "p97-ATX3-RNF8-(proteasome) complex" (pp. 8, 18), since the co-IPs could equally well reflect binary interactions.

Answer 10: We have a physical complex between ATX3-VMB (that cannot bind p97) and RNF8 together with the main p97 cofactor complex (Ufd1-Npl4) (Fig. 3E). This suggests that ATX3-RNF8 and Npl4-Ufd1 form a complex without p97. Next, the reciprocal Co-IP with p97-wild type also detected ATX3 and RNF8 in the complex and this complex is not disrupted in ATX3-knock out cells (Fig. 3F). Altogether, this suggests that ATX3 and p97 can interact with RNF8 independently of each other. Further, RNF8 hyper-accumulates in Co-IP with p97 ATPase inactive variant (p97-EQ) in ATX3-knockout cells (Fig. 3F) and ATX3-VBM variant still interacts with RNF8 and Npl4-Ufd1 (the main p97 cofactors). Altogether this strongly suggests that we have one physical complex between p97-ATX3 and RNF8.

3. Figs. 1F, 2AC, 4BC, 6C: The conditions used for the denaturing IPs (1% SDS) will certainly denature most if not all proteins in the lysates, but will not prevent subsequent unspecific interactions of "sticky" proteins, also including (partially) denatured ubiquitin chains. This leaves the possibility that not all ubiquitinated species detected in these experiments are RNF8 conjugates. Key findings need to be confirmed using His-Ub expressing cells in fully denaturing (GdnHCl/urea) Ni-NTA pull-down experiments, preferably detecting endogenous RNF8 and H1, respectively. Regarding Fig. 6C: Was Flag-H1 isolated from total cell lysates or from a chromatin-enriched fraction? The latter would be the appropriate experiment.

Answer 11: We think that referee 3 overlooked the fact that our denaturing protocol is not only done with 1% SDS but also with heat denaturation (95°C/5 min; please see a Material and Methods section, page 37). Also, the RNF8 precipitates before elution were 3X washed with 1M NaCl and 1% Triton X-100. Unfortunately, washings with 1M NaCl and 1% Triton X-100 was not mentioned in the Materials and Methods section, but it will be added. Similar denaturing protocols were used elsewhere (El-Shemerly et al., Cancer Res. 2005, Ramadan et al. Nature 2007, Wimmer et al. EMBO Rep. 2008).

Under these stringent conditions there is a complete protein denaturation similar to the denaturing protocol suggested by this referee.

4. The proteomics data summarized in Fig. 3A and the corresponding experimental methods have not been subject to peer review. Consequently, the authors should either show all necessary details and results, or remove the figure and just refer to the data as unpublished results in the text.

Answer 12: This is ongoing project in our lab and we cannot disclose all mass-spectrometry results on the p97-protein interactome. However, we will add the raw SILAC data for Figure 3 and explain in the Materials and Methods section how the mass-spectrometry was performed and analysed.

5. NMS-873 does not disrupt the hexameric state of p97 (p. 9).

Answer 13: Thank you for pointing this out. It will be corrected in the text by stating that NMS-873 affects distortion of interprotomer communication between D1 and D2 domains and thus impair p97 enzymatic activity.

Thank you for your response letter answering to the comments of our three arbitrating referees. After having considered them in detail, I appreciate your clarifications and plans for revising the manuscript to address the reviewers' points. In particular, I understand that further dissecting the involvement and regulation of distinct ubiquitin chain types, or the testing of effects on various RNF8 downstream targets, would go beyond the scope of revision of this study. Points that will be important for a revised version are the incorporation of Npl4-Ufd1 inactivation analyses, more decisive ATX3-MDC1 co-IP data, and better description of the proteomics approach, as outlined in your response letter.

We would like to thank the referees on their comments and suggestions on our manuscript entitled "**Homeostasis of the E3-ubiquitin ligase RNF8 by the p97-Ataxin 3 complex preserves genome stability by regulating DNA repair pathway choice**" (EMBO J-2019-102361).

We are very happy to hear that referees 1 and 2 are completely satisfied with our revised version of the manuscript prepared for another journal. However, referee 3, although convinced and satisfied with our manuscript, raises some specific issues. We believe we addressed referee 3's concerns, either by clarifying certain aspects in the text, or by performing additional experiments.

Specifically,

Referee#1

You asked me to look over this manuscript and its review process with regard to the previous Dantuma paper (also on ATX3 and the DDR). I really think the Ramadan group have done all they reasonably could to tackle the discrepancy with the previous report - including asking another lab to look at it.

Why the difference exists I'm not sure - but its worth remembering that the impact Dantuma reported on MDC1 was slight and needed sophisticated analysis. As regard answering the reviewers again I think they are there. While some minor elements might be nice to see (RNF8 foci time course for example on ATX depletion in G1 Vs S/G2 cells) I doubt the story over-all would be much improved.

Answer 1: Thank you for your support of our manuscript.

Referee#2

I was asked to comment on the manuscript and prior response to reviews. Overall, this is an interesting study and helps clarify differences between RNF8 and RNF168. It also has some murky areas, which is perhaps not unexpected given the myriad processes that p97 is involved with. I am generally convinced by the experimental evidence, however think that it would be valuable to use more knockout lines rather than double siRNA to perform epistasis experiments. These are less convincing and could be performed using better tools.

Answer 2: Thank you for your support of our manuscript. We highly appreciate your opinion and the suggestion to use more knockout lines. We also believe that ATX3 and RNF8 epistasis would be much better to analyse in double ATX3/RNF8 knockout cells. However, RNF8 is a fitness gene in all tested human cell lines. Thus, human cells would not survive RNF8 knockout, or cells would terminally adapt to the loss of RNF8, which could mask the observed epistatic phenotype.

Referee#3

The manuscript by Singh et al. contains an enormous amount of data, and this reviewer appreciates its difficult history in light of a competing publication by the Dantuma lab (Pfeiffer et al, 2017). However, this reviewer also shares the criticism raised by previous reviewers that the manuscript combines two loosely connected stories about the control of RNF8 by p97 and ATX3 that are difficult to reconcile. The first part reports that ATX3 antagonizes the p97-dependent proteasomal degradation of RNF8. This could simply reflect a house-keeping quality control function of p97 and ATX3 and, as such, would not be particularly original given a number of previous, similar observations for other p97-ATX3 substrates. In order

to make this part suitable for publication in EMBOJ, a significant number of additional experiments leading to substantial mechanistic insights into the role of ATX3 in p97-mediated substrate turnover would be required (probably also including a reconstituted in vitro assay), in addition to consolidating the existing data (see specific points below). Alternatively, the first part could be largely or completely omitted, and the manuscript could be re-focused on the much more interesting second part, i.e. the role of p97 and ATX3 in controlling RNF8 level and activity at the sites of DSBs.

Answer 3: Thank you for your opinion and the suggestions. As discussed at several places throughout the manuscript, the regulation of RNF8 homeostasis at the protein level is essential for tumorigenesis, cancer cell proliferation and facilitates cancer chemoresistance and poor patient prognosis (Lee HJ et al., Mol. Cell, 2016, Wang S et al., Biochim Biophys Acta Mol Basis Dis. 2017, Li L et al., J Clin Invest. 2018). These reports suggest that either high or low RNF8 protein expression leads to cancer development and poor patient prognosis.

Also, a recent report nicely demonstrates the role of RNF8 in DNA replication (Schmid et al., Mol Cell, 2018). However, how RNF8, one of the human fitness genes, is regulated at the protein level is not known. Thus, we believe our discovery of the p97-ATX3 complex as one of the main regulators of RNF8 homeostasis under unchallenged/physiological conditions is essential for both DNA replication and cancer.

We agree that this part of the manuscript (RNF8 regulation under unchallenged conditions) could be a separate manuscript and it was highly appreciated by the original reviewer 3 from another journal. However, we believe it is important to report that the p97-ATX3 complex is the main regulator of RNF8 homeostasis under both unchallenged and genotoxic conditions in one place. This is the main message of this manuscript.

...This second part contains a comprehensive and for the most part highly convincing description of the involvement of ATX3 in DSB repair by NHEJ, but not HR, and thus in repair pathway choice. This story should still be strengthened in two ways: First, the exact roles of ATX3 and RNF8 in the process should be discussed in more detail. E.g., is RNF8 at DSBs auto-ubiquitylated with K63 or K48 chains? (The accumulation of RNF8 and K63-Ub, but not K48-Ub upon depletion of ATX3 would suggest the first possibility.)

Answer 4: We showed that an RNF8 catalytically inactive variant (RING finger mutation in C403S) hyper-accumulates at sites of DNA damage (Fig EV3 G and H) suggesting that RNF8 without its own E3-ligase activity cannot be removed. Furthermore, depletion of ATX3 results in increased K63-ubiquitin signal on chromatin, and this is completely abolished by RNF8 co-depletion (Fig EV3 E and F). In contrast, the p97-ATX3 complex regulates preferentially the K48-ubiquitin signal on RNF8 (Fig 2A, 4D and E). RNF8 is preferentially ubiquitinated by K48-ubiquitin chains in both soluble and chromatin fractions (Fig EV1D). K63-ubiquitination of RNF8 is also observed on both soluble and chromatin fractions but to a much lesser extent when compared to K48-ubiquitination. However, the affinity of specific K48- or K63-ubiquitin chain antibodies are similar (Fig EV1E).

Thus, as discussed in our manuscript, we do not yet understand why ATX3 has different preferences in the regulation of soluble vs chromatin bound RNF8. Different post-translational modifications on ATX3 after IR (such as

phosphorylation) might regulate ATX3 preferences for either soluble or chromatin-bound RNF8. For example, ATX3 has a SUMO-Interacting Motif (SIM) and it has been shown that ATX3 is recruited to the sites of DNA lesions by SUMO (Pfeiffer et al. EMBO J, 2017). Therefore, it is possible that chromatin-bound RNF8 is also SUMOylated, and both ubiquitin and SUMO interactions between ATX3 and RNF8 regulate the different outcomes of RNF8 in chromatin vs. soluble fractions. These important questions could not be addressed in this already very complex manuscript as it would be necessary to do more thorough research.

Why does the accumulation of RNF8 upon depletion of ATX3 result in a counterintuitively reduced ubiquitination of its substrate, histone H1? Why/how would other substrates "suppress the timely ubiquitination of H1" (p. 15) when its E3 ligase RNF8 is present in excess?

Answer 5: We discussed this question on page 21 (discussion part). Also, recent findings suggest that histone H1 might not be the main substrate of RNF8, but L3MBTL2 (Nowsheen S, Nat Cell Biology 2018). It seems there is still an open question in the field: what is the main substrate of RNF8 at sites of DNA damage that is important for propagation of the ubiquitin signal? Due to its different focus - *regulation of RNF8 by the p97-ATX3 system* – this manuscript could not address this important question.

What is the role of UFD1-NPL4 in RNF8 turnover at DSBs? (A potential accumulation of RNF8 upon depletion of UFD1 or NPL4 was not tested.)

Answer 6: We depleted UFD1 or NPL4 with siRNAs and tested RNF8 turnover at the sites of two-photon laser or IR-induced DNA damage in living or fixed cells, respectively (Fig EV2K, L and N). Depletion efficacy was confirmed by Western blot (Fig EV2M). Depletion of Npl4 affects Ufd1 stability in U2OS cells and vice versa, suggesting that the Npl4-Ufd1 heterodimer works as a single complex. Inactivation of the Npl4-Ufd1 complex did not significantly affect RNF8 turnover at the sites of NIR-multiphoton laser induced DNA damage in living cells. Inactivation of ATX3 by siRNA or chemical inhibition of p97 by CB-5083 served as positive controls for the hyper-accumulation of RNF8 at the sites of DNA lesions. A similar result was also observed when RNF8 accumulation was analysed at the sites of IR-induced DNA damage in fixed cells.

Altogether, these data suggest that the Npl4-Ufd1 complex does not have a strong effect on RNF8 accumulation at the sites of DNA damage and that the p97-ATX3 complex might potentially work without the Npl4-Ufd1 complex for RNF8 removal at the sites of DNA damage.

Second, the key hypothesis that H1 ubiquitination is impaired because of loosened chromatin and/or dissociation of H1 from the vicinity of DSBs should be confirmed experimentally, e.g. by H1 and CHD4 staining at DSBs by IF or by appropriate PLA experiments. (The Figure B for reviewers shows total chromatin of unstressed cells, which may be irrelevant for the vicinity of DSBs. Also, the data are not really convincing, as unequal total amounts of chromatin appear to be loaded in lane 2 vs 5.)

Answer 7: Impaired H1-ubiquitination in ATX3-inactivated cells is not the key hypothesis of this manuscript, but that the p97-ATX3 complex regulates the main

and first E3-ubiquitin ligase RNF8 in DNA damage response. How RNF8 regulates other substrates is not well understood in the literature and further work will be required (see answer 5). Unfortunately, this manuscript could not address this question. Thus, in this already complex manuscript that contains an enormous amount of data (stated by this referee), we cannot additionally analyse how the ATX3-RNF8 complex regulates chromatin status at the sites of DNA damage.

Regarding the results contradicting the previous paper by Pfeiffer et al, the authors present convincing microscopy data supporting their conclusions. However, as to the question if MDC1 is a target/interactor of ATX3, the negative result in Fig. S4G is not really convincing, given that the IP of GFP-ATX3 is very inefficient (depletion rather than enrichment in comparison to the input levels). To strengthen their point, the authors should either provide a better IP of GFP-ATX3, or probe the Flag-IPs in Fig. 3D with a MDC1 antibody.

Answer 8: We did this experiment multiple times (4 independent experiments) and we now provide another GFP-ATX3 co-immunoprecipitation blot where ATX3 has efficiently been isolated (Fig EV3A). In each of our 4 independent GFP-ATX3 Co-IP experiments MDC1 had never been isolated in the complex with ATX3, but RNF8 was.

Moreover, it might be advantageous to discuss all consensus and discrepancies with Pfeiffer et al in one paragraph of the Discussion section, rather than spreading it over several places.

Answer 9: We moved a large part of discussion on the discrepancies of our results with the Pfeiffer et al. manuscript to the Discussion section.

Specific points:

1. While the CHX chase assays monitor the degradation of endogenous RNF8, all experiments in Figs. 1 through 4 addressing the ubiquitination state of RNF8 and its interaction with p97 and ATX3 were performed with cells strongly overexpressing RNF8, p97 and/or ATX3. Since overexpression can lead to completely erroneous conclusions, at least the key results must be confirmed at endogenous expression levels. For instance, the HA blots of the inputs in Figs. 2A and 2C show that overexpression of FLAGRNF8 leads to a massive increase of total (!) ubiquitinated proteins in the lysate, which is a completely artificial setting.

Answer 10: The rescue of endogenous RNF8 degradation in ATX3-depleted cells with either ATX3-wt expression but not with ATX3-VBM variant (p97-binding deficient) (Fig 5E and F) or by proteasome inhibitor MG132 (Fig 4C), suggests that endogenous RNF8 is regulated by the ubiquitin signal. On top of it, we clearly demonstrated that over-expressed RNF8wt but not its E3-ligase defective variant (RNF8-RING) is prone to degradation (Fig 2E, F and compare with Fig 2B and C). Altogether, these results suggest that endogenous RNF8 is degraded in a ubiquitin-dependent manner (proteasome and p97 -dependent degradation) and that overexpressed RNF8 follows similar degradation rules and kinetics as endogenous RNF8.

I agree, this could be confirmed by isolating endogenous RNF8 and then checking the ubiquitination status of RNF8. However, the majority of published manuscripts,

when analysing the ubiquitination status of specific protein, use the over-expressed protein as a read out - as we did here for RNF8.

In summary, as we demonstrated that endogenous RNF8 degradation is similar to exogenous RNF8 degradation and this process depends on the proteasome, p97 and the E3-ligase activity of RNF8, we think it is not necessary to additionally demonstrate that endogenous RNF8 is also ubiquitinated.

2. Please note that the co-IP results do not support the existence of a "p97-ATX3-RNF8-(proteasome) complex" (pp. 8, 18), since the co-IPs could equally well reflect binary interactions.

Answer 11: Our Co-IP results are additionally supported by the functional experiments where it is clear that p97, ATX3 and RNF8 work together as one functional and physical complex.

3. Figs. 1F, 2AC, 4BC, 6C: The conditions used for the denaturing IPs (1% SDS) will certainly denature most if not all proteins in the lysates, but will not prevent subsequent unspecific interactions of "sticky" proteins, also including (partially) denatured ubiquitin chains. This leaves the possibility that not all ubiquitinated species detected in these experiments are RNF8 conjugates. Key findings need to be confirmed using His-Ub expressing cells in fully denaturing (GdnHCl/urea) Ni-NTA pull-down experiments, preferably detecting endogenous RNF8 and H1, respectively. Regarding Fig. 6C: Was Flag-H1 isolated from total cell lysates or from a chromatin-enriched fraction? The latter would be the appropriate experiment.

Answer 12: We think that referee 3 overlooked the fact that our denaturing protocol is not only done with 1% SDS but also with heat denaturation (95°C/5 min; please see Materials and Methods section, page 33). Also, the RNF8 precipitates before elution were washed with 1M NaCl and 1% Triton X-100. Unfortunately, the washing step with 1M NaCl and 1% Triton X-100 was not mentioned in the Materials and Methods section, but it is now added. Similar denaturing protocols were used elsewhere (El-Shemerly et al., Cancer Res. 2005, Ramadan et al. Nature 2007, Wimmer et al. EMBO Rep. 2008). Under these stringent conditions there is a complete protein denaturation similar to the denaturing protocol suggested by this referee.

4. The proteomics data summarized in Fig. 3A and the corresponding experimental methods have not been subject to peer review. Consequently, the authors should either show all necessary details and results, or remove the figure and just refer to the data as unpublished results in the text.

Answer 13: We added the raw SILAC data (Table EV1) for Figure 3A and explained in the Materials and Methods section how the mass-spectrometry was performed and analysed.

5. NMS-873 does not disrupt the hexameric state of p97 (p. 10).

Answer 14: Thank you for pointing this out. We corrected this in the text by stating that NMS-873 affects distortion of interprotomer communication between D1 and D2 domains and thus impair p97 enzymatic activity.

2nd Editorial Decision

19th September 2019

Thank you for submitting your revision for our consideration. I have now gone through the revised manuscript and assessed your responses to the referee reports, and I am pleased to say that I see no more objections from the scientific side towards publication in The EMBO Journal.

Corresponding Author Name: Kristijan Ramadan

Journal Submitted to: EMBO J

Manuscript Number: EMBO J-2019-102361